# Role of flying cars in sustainable mobility

Akshat Kasliwal [1,2], Noah J. Furbush[1,3], James H. Gawron [2], James R. McBride[1], Timothy J. Wallington [1], Robert D. De Kleine [1], Hyung Chul Kim [1] & Gregory A. Keoleian [2]

Interest and investment in electric vertical takeoff and landing aircraft (VTOLs), commonly known as flying cars, have grown significantly. However, their sustainability implications are unclear. We report a physics-based analysis of primary energy and greenhouse gas (GHG) emissions of VTOLs vs. ground-based cars. Tilt-rotor/duct/wing VTOLs are efficient when cruising but consume substantial energy for takeoff and climb; hence, their burdens depend critically on trip distance. For our base case, traveling 100 km (point-to-point) with one pilot in a VTOL results in well-to-wing/wheel GHG emissions that are 35% lower but 28% higher than a one-occupant internal combustion engine vehicle (ICEV) and battery electric vehicle (BEV), respectively. Comparing fully loaded VTOLs (three passengers) with ground-based cars with an average occupancy of 1.54, VTOL GHG emissions per passenger-kilometer are 52% lower than ICEVs and 6% lower than BEVs. VTOLs offer fast, predictable transportation and could have a niche role in sustainable mobility.

[1] Research and Innovation Center, Ford Motor Company, Dearborn, Michigan 48121, USA. [2] Center for Sustainable Systems, School for Environment and Sustainability, University of Michigan, 440 Church Street, Ann Arbor, Michigan 48109, USA. [3] Department of Aerospace Engineering, University of Michigan, 1320 Beal Avenue, Ann Arbor, Michigan 48109, USA. Correspondence and requests for materials should be addressed to G.A.K. (email: gregak@umich.edu)

The transportation sector faces the challenge of meeting growing demand for convenient passenger mobility while reducing congestion, improving safety, and mitigating emissions. Automated driving and electrification are disruptive technologies that may contribute to these goals, but they are limited by congestion on existing roadways and land-use constraints. Electric vertical takeoff and landing aircraft (VTOLs) could overcome these limitations by enabling urban and regional aerial travel services. VTOLs with tilt-rotor, duct, and wing designs, such as the GL-10 prototype designed by NASA[1], combine the convenience of local takeoff and landing like a helicopter with the efficient aerodynamic flight of an airplane. Although smaller and larger designs are possible, several companies are considering craft that can carry four to five occupants[2]. Initially, these VTOLs would likely be piloted taxi services, but with advances in aviation regulation and sensor and processor technology, could transition toward future automated control[3].

Electrification is a propulsion strategy for improving the sustainability of both aerial and ground-based transportation modes, owing to the superior efficiency of electric powertrains compared with combustion engines. One critical efficiency enabler for VTOLs is distributed electric propulsion (DEP), which uses physically smaller, electrically-driven propulsors. These propulsors can be used with greater flexibility to leverage the benefits of aero-propulsive coupling and improve performance compared with more traditional designs[4]. This enables aerodynamically optimized designs, such as articulating propellers and high aspect-ratio blown wings, which allow efficient VTOL energy performance and significant noise reduction. DEP could facilitate VTOL success in the urban aerial taxi space, where conventional helicopters or vertical-lift aircraft have struggled.

In principle, VTOLs can travel the shortest distance between two points, and their relatively modest sizes would enable near point-to-point service. Conversely, road networks are much less direct and consequently have an associated circuity factor, defined as the ratio of the shortest network route to the Euclidian distance between two points[5]. This benefit of VTOL aerial systems could favor energy and travel-time performance, particularly in locations with congested and circuitous routing. High VTOL cruise speeds could reduce travel time further. Significant time savings and associated productivity gains could be a key factor in consumer adoption of VTOL transportation.

There are many questions that need to be addressed to assess the viability of VTOLs including cost, noise, and societal and consumer acceptance. Our analysis assesses the environmental sustainability of VTOLs compared with ground-based passenger cars. There have been few studies of VTOLs' potential climate change implications[6,7].

We report the first comprehensive assessment of the primary energy and GHG emissions impacts of using electric VTOLs vs. ground-based light-duty vehicles for passenger mobility. Our analysis first focuses on a vehicle-to-vehicle comparison with one occupant (i.e., the pilot or driver) traveling point-to-point distances ranging from 5 to 250 km. The base case is assessed for a 100 km distance. As part of a sensitivity analysis, we compare the results on a passenger-kilometer traveled (PKT) basis. The VTOL is assumed to have three passengers and one pilot (i.e., four occupants), as it will most likely be used in a transportation-as-a-service business model where service providers seek to maximize utilization rates. Ground-based cars are assumed to be personally owned with a typical loading of 1.54 passengers/occupants[8]. Modeling details are available in the Methods section, with uncertainties explored in the Sensitivity Analysis. We assess use-phase burdens associated with both aerial (well-to-wing) and ground-based (well-to-wheel) transport. Total fuel cycle impacts encompass both upstream (mining, refining, and transportation of the fuel source) and downstream (operational) activities. Burdens from other life cycle stages, such as vehicle production and end-of-life, are not considered owing to a lack of standardization in VTOL fabrication materials, manufacturing processes, and design specifications.

To quantify the use-phase sustainability of these mobility systems, two key metrics are chosen: primary energy use in units of megajoules [MJ] and GHG emissions in units of kilograms of carbon dioxide equivalents [kg-$CO_2$e] on a 100-year global warming potential basis. Subsequently, differences in real-world occupancies are explored by normalizing those metrics on a PKT basis, which is useful when comparing different passenger transport modes[9]. We also compare the travel time of VTOLs vs. cars. Piloted operation for both modes of mobility is the basis of our analysis. Connected and automated operation are beyond the existing scope and will be considered in future work.

We find that for our base case with 100 km point-to-point trips, VTOL GHG emissions are 35% lower than internal combustion engine vehicles (ICEVs), but 28% higher than battery electric vehicles (BEVs). Normalizing base-case emissions per PKT with expected loading gives VTOL burdens (with three passengers) that are 52% and 6% lower than for the ICEV and BEV (with 1.54 passengers), respectively. For short trips (up to 35 km), which dominate trip frequency for conventional cars, VTOLs have higher energy consumption and GHG emissions than ground-based vehicles. Time savings for VTOL rides compared with cars (83% for a 100 km trip) could act as a driver for consumer adoption. From the viewpoint of energy use and hence GHG emissions, it appears that VTOLs could have a niche role in sustainable mobility, particularly in regions with circuitous routes and/or high congestion.

## Results

**VTOL results**. The flight profile shown in Fig. 1 is broken up into the following five phases: takeoff hover, climb, cruise, descent, and landing hover. VTOLs, such as the NASA GL-10 depicted in Fig. 2, have a different travel time, speed, and power consumption profile during each phase, as discussed in the Methods section. The base-case GHG emissions associated with each phase are shown in Fig. 3 for trips between 5 and 250 km. A minimum of 5 km was chosen due to the 2.5 km horizontal slant range during both climb and descent phases. The base-case scenario of transporting one occupant over a point-to-point distance of 100 km has GHG emissions of 15.7 kg-$CO_2$e. For shorter travel distances, where energetically intensive hover dominates the flight profile, the VTOL compares less favorably than it does for longer distances, where efficient cruise dominates the flight profile. Primary energy results follow the same trends as GHG emissions (refer to Supplementary Figs. 1 and 2).

**Ground-based vehicle results**. On-road adjusted city and highway fuel economies for the BEV were 304 Wh mi$^{-1}$

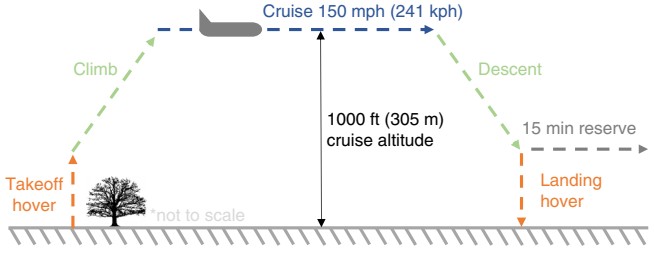

**Fig. 1** VTOL flight profile. The five phases of VTOL travel are takeoff hover, climb, cruise, descent, and landing hover. Each phase will have a different travel time, velocity, and power consumption

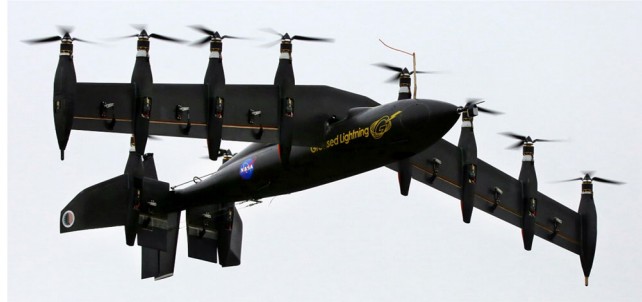

**Fig. 2** NASA GL-10 VTOL[1]. The takeoff and landing hover configuration for a prototype NASA VTOL is shown here. The tilt rotor and wing design combines the convenience of local takeoff and landing like a helicopter with the efficient aerodynamic flight of an airplane

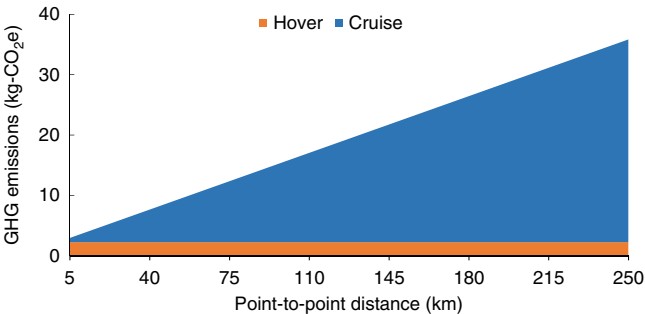

**Fig. 3** VTOL GHG emissions over a range of trip distances. The GHG emission results for a single-occupant VTOL are broken out by the hover and cruise phases over trip distances from 5 to 250 km. The climb phase is modeled as part of cruise. Furthermore, the takeoff and landing hover phases are combined for simplicity and the powerless descent phase is omitted, as it is assumed to have zero emissions. See the Methods section for details

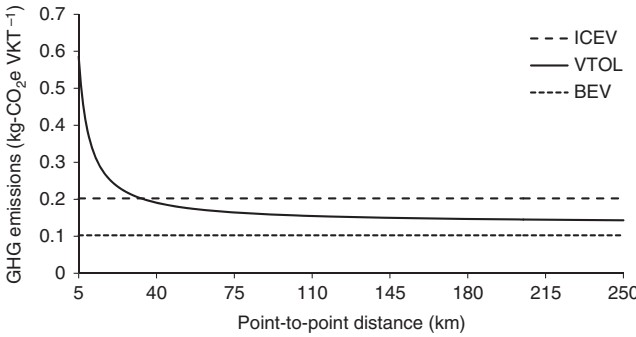

**Fig. 4** GHG emissions normalized by vehicle-kilometers traveled. The GHG emission results for single-occupant VTOLs and ground-based vehicles (ICEV and BEV) are normalized by vehicle-kilometers traveled (VKT). This illustrates the impact of amortizing the fixed burden from the hover phase over longer distances. The VTOL GHG emissions break even with the ICEV at 35 km

(109.2 MPGe) and 309 Wh mi$^{-1}$ (107.5 MPGe), respectively[10]. This results in a fuel economy of 108.5 MPGe for a combined 55% city/45% highway driving cycle[11]. For the ICEV, on-road adjusted city and highway fuel economies were 30.7 and 39.5 MPG, respectively, yielding a combined fuel economy of 34.1 MPG[10,11]. See Methods section for details. Accounting for effects from fuel carbon intensity and a circuity factor of 1.20[5] (to incorporate actual road distance traveled between origin and destination) yields use-phase emissions values for the base-case

single-occupant scenario of 12.3 kg-CO$_2$e for the BEV and 24.3 kg-CO$_2$e for the ICEV. The lower burdens from the BEV reflect the higher overall system efficiencies of electrified platforms over internal-combustion platforms. The fuel-to-motion conversion efficiency of ICEVs is 12–30%, depending on the drive cycle, whereas for BEVs this efficiency is 72–94%[12].

**VTOL vs. ground-based vehicle comparison**. Figure 4 compares VTOL vs. ICEV and BEV emission intensity (kg-CO$_2$e VKT$^{-1}$) as a function of point-to-point trip distance (vehicle-kilometers traveled (VKT)) for our base case with one occupant in each vehicle. The associated primary energy used is shown in Supplementary Fig. 3. ICEV and BEV base-case emissions were found to be roughly 0.20 and 0.10 kg-CO$_2$e VKT$^{-1}$, respectively. As indicated in Fig. 3, VTOLs incur significant emissions for hover but are efficient in cruise. As a result, the VTOL emissions per VKT shown in Fig. 4 are ~0.59 kg-CO$_2$e VKT$^{-1}$ for the shortest trip (5 km) but decrease rapidly with increasing trip length, tending toward an asymptotic value of ~0.14 kg-CO$_2$e VKT$^{-1}$ for a 250 km trip. Base-case VTOL emissions (for the 100 km trip) are 0.15 kg-CO$_2$e VKT$^{-1}$.

The ICEV performs better than the VTOL up to ~35 km, where aerial flight is dominated by the energy-intensive hover mode. GHG emissions for VTOLs drop substantially below those from ICEVs for trips longer than ~50 km. For long-distance trips, VTOLs can leverage efficient cruise performance to outperform ICEVs. The VTOL emissions approach, but do not match, those from BEVs for distances > ~120 km. For our base-case travel distance of 100 km (point-to-point), the VTOL has well-to-wing/wheel GHG emissions that are 35% lower but 28% higher than the ICEV and BEV, respectively.

Initially, VTOLs are likely to operate as aerial taxis, and service providers would target near-full occupancy from a utilization–maximization standpoint, similar to current commercial airlines. Passengers could be incentivized to share VTOL rides given the significant expected time savings from flying. Thus, it seems likely that the average occupancy of VTOLs will be greater than conventional passenger cars. Given an expected occupancy difference, it can be argued that the emission burdens between VTOLs and ground-based vehicles should be compared on a PKT basis rather than the VKT basis shown in Fig. 4. The results of this assessment are described in the Sensitivity Analysis.

**Sensitivity analysis**. Extensive variability exists in VTOL design and operational domains. The sensitivity analysis presented here includes the variation of six key VTOL parameters from the base-case values (for 100 km point-to-point travel). Table 1 contains the definitions for the key parameter input values and associated sources. Figure 5a, b summarize the results of the analysis. Figure 5a shows the sensitivity of the base-case VTOL emissions (kg-CO$_2$e VKT$^{-1}$) to grid carbon intensity, wind, lift-to-drag ratio ($L/D$), battery-specific energy, and powertrain assumptions.

First, changing the 2020 electrical grid carbon intensity from the US average mix to the California and Central-and-Southern Plains grids results in a 52% decrease and 41% increase in emissions, respectively. It is noteworthy that a similar effect will be seen with the BEV when comparing these VTOL results with the BEV baseline. Second, although impacts of wind could equalize fluctuations in emissions for a defined route over multiple iterations of travel in an aerial taxi service, weather remains an important consideration that can affect VTOL energy use for a given trip. We estimate a 16% reduction in base-case emissions with a favorable 30-knot tailwind. Conversely, a

**Table 1 VTOL modeling input parameters**

| Parameter | Notation | Min value | Base-case Value | Max value | Unit | Source |
|---|---|---|---|---|---|---|
| Battery energy capacity | $C$ | — | 140 | — | kWh | Uber[7] |
| Battery depth of discharge | DoD | — | 0.8 | — | — | Harish et al.[23] |
| Battery-specific energy | $E^*$ | 250[a] | 400[b] | — | Wh kg$^{-1}$ | a: Elgowainy et al.[10] |
| | | | | | | b: Misra[37] |
| Gravitational acceleration | $g$ | — | 9.81 | — | m s$^{-2}$ | Constant |
| Cruise altitude | $h$ | — | 1000 | — | ft | Uber[3] |
| | | | (305) | | (m) | |
| Cruise lift-to-drag ratio | $L/D$ | 13 | 17 | 20 | — | Uber[7] |
| Takeoff mass | $m$ | — | 1187.5 | 1660 | kg | Calculated and Datta[2] |
| Battery mass | $m_b$ | — | 350 | 560 | kg | Calculated |
| Payload mass | $m_p$ | — | 87.5 | 350 | kg | FAA[38] |
| Structural mass | $m_s$ | — | 750 | — | kg | Calculated and Ullman et al.[6] |
| Rate of climb | ROC | — | 1000 (5) | — | fpm (m s$^{-1}$) | Stoll and Mikic[35] |
| Rate of descent | ROD | — | 1000 (5) | — | fpm (m s$^{-1}$) | Stoll and Mikic[35] |
| Cruise true airspeed | $V$ | — | 150 (241.4) | — | mph (kph) | Uber[3] |
| Wind speed | $V_{wind}$ | −30 | 0 | 30 | Knots | Chosen input |
| Battery mass fraction | $\beta$ | — | 0.23 | 0.33 | — | Calculated |
| Disk loading | $\delta$ | — | 450 | — | N m$^{-2}$ | Stoll[32] |
| Climb and cruise system efficiency | $\eta_c$ | 0.700[a] | 0.765[b] | 0.800 | — | a: Stolaroff et al.[25] |
| | | | | | | b: Brown and Harris[26] |
| Hover system efficiency | $\eta_h$ | — | 0.63 | — | — | Ullman et al.[6] |
| Battery charge-discharge efficiency | — | — | 0.90 | — | — | Cooney et al.[28] |
| Primary-to-delivered electricity efficiency | — | — | 0.408 | — | — | GREET[29] |
| Electrical grid carbon intensity | — | 0.065 (CA) | 0.135 (US) | 0.190 (Central) | kg-CO$_2$e MJ$^{-1}$ | GREET[29] |
| Sea-level air density | $\rho$ | — | 1.225 | — | kg m$^{-3}$ | Engineering toolbox[39] |

30-knot headwind increases these emissions by nearly 26%. Third, we examine the $L/D$ ratio. An upper-bound aerodynamic efficiency value of 20 during cruise would reduce base-case GHG emissions by almost 13%. Conversely, a lower bound for the cruise $L/D$ of 13 will result in a 26% increase from the baseline emissions. Fourth, we consider battery-specific energy. If the specific energy is reduced from 400 Wh kg$^{-1}$ in the VTOL baseline to the 250 Wh kg$^{-1}$ assumed for the BEV[10] (while keeping the battery capacity constant at 140 kWh), the emissions will increase by 18%. This also reduces the safe operating range from 250 to 220 km due to the higher energy consumption required to account for the added battery weight. Fifth, system efficiency is another important variable to consider. The efficiency is driven by the electric powertrain and modern propeller designs for lift and cruise functionality. This is bounded by a value of 70% on the lower end and 80% on the upper end. These result in a nearly 8% increase and 4% decrease from the baseline emissions, respectively.

Sixth, we consider the impact of passenger loading on emissions calculated on a PKT basis. As noted previously, it seems likely that the average VTOL loading will be higher than for conventional ground-based vehicles. There are no available empirical data upon which to assume a typical VTOL occupancy; hence, we consider one to three passengers (alongside a pilot) spanning the complete range for the craft considered (corresponding to a maximum payload of 350 kg). The average number of passengers in a ground vehicle is 1.54 (including the driver)[8], which forms a reasonable basis of comparison with the VTOL. Figure 5b shows the PKT results for VTOL vs. an ICEV and BEV, noting that the pilot in the VTOL is not considered a passenger. We define an occupant as any person who is physically contained in the vehicle, whereas a passenger is an occupant for whom the trip is being made. Therefore, the VTOL pilot is not considered a passenger. As seen from Fig. 5b, for two or more passengers the VTOL outperforms the ICEV and for three passengers the VTOL outperforms the BEV on a PKT basis. Specifically, a three-passenger VTOL has burdens that are 52% and 6% lower than for

a 1.54-passenger ICEV and BEV, respectively. Figure 5b indicates that VTOLs operating at or near-full capacity are relatively efficient, outperforming average-occupancy BEVs in the base case. Supplementary Fig. 4 contains the sensitivity analysis results for the BEV and ICEV.

**Travel time**. The point-to-point VTOL flight path results in a 100 km trip time of about 27 min, with a cruise speed of 150 mph (roughly 241 kph). For a highly congested commute of a similar distance, approximately the span of a major city, time savings of point-to-point travel can be significant. VTOL travel time is dependent on many factors that are hard to collectively characterize, including air traffic and airspace restrictions. Weather challenges are inherent with aircraft operation, which can create travel-time variability. Thirty-knot headwinds and tailwinds are considered as bounds in the Sensitivity Analysis, representing inclement weather that is potentially still safe for flight. This results in a nearly 3 min increase or decrease in travel time for the base-case 100 km trip. Although travel time for VTOLs can vary with weather conditions, the variability is relatively small and can be predicted given reasonable weather forecasts. Predictability is a major advantage of VTOL mobility, particularly in locations where road systems are highly congested and ground travel times highly unpredictable.

On the ground, adopting the five-cycle test procedure (see Methods section) yields an average speed of 20.6 mph (33.2 kph) for all-city driving and 58.5 mph (94.1 kph) for all-highway driving. This results in an average speed of 29.1 mph (46.8 kph) for a combined 55% city/45% highway driving cycle. Assuming an average circuity factor of 1.20, defined as the ratio of actual and straight-line distance, leads to a travel time of 154 min for the base case. For context, a trip of similar length from Irvine to Malibu can take between 120 and 210 min during rush hour according to a Google Maps estimate[13]. For ground-based vehicles, variability in travel time is significant. From the 55% city/45% highway base case, the travel time increases by 41%

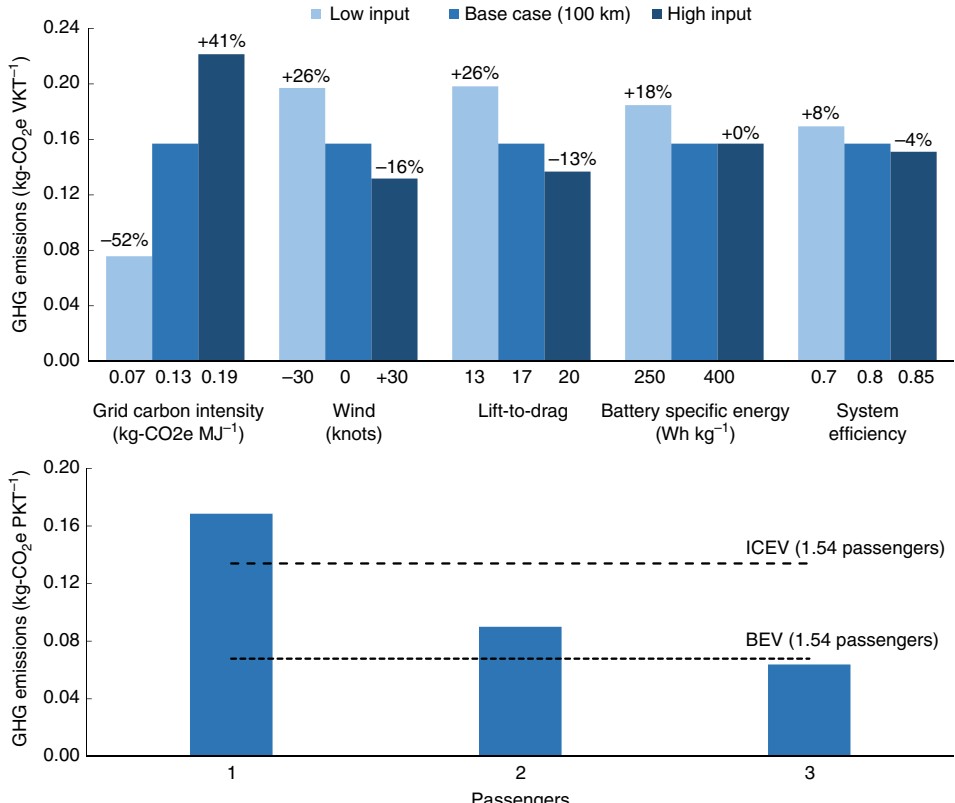

**Fig. 5 a** Sensitivity analysis for VTOL base-case scenario. Five key modeling parameters are individually varied over realistic bounded ranges within the modeling of the 100 km base-case VTOL scenario. Variation in the electrical grid carbon intensity has the largest impact on the results, whereas the range of system efficiencies show the smallest change. **b** Sensitivity analysis for passenger loading. The 100 km base-case VTOL scenario is modeled with passenger loading varying from 1 to 3. GHG emission results are normalized by personal-kilometers traveled (PKT) to illustrate the impact of allocating the burden over more travelers. Dashed horizontal lines indicate results for ground-based vehicles (ICEV and BEV) with an average occupancy of 1.54 passengers. It should be noted that the pilot is not considered a passenger in the VTOL, whereas the driver is considered a passenger in the ground-based cars

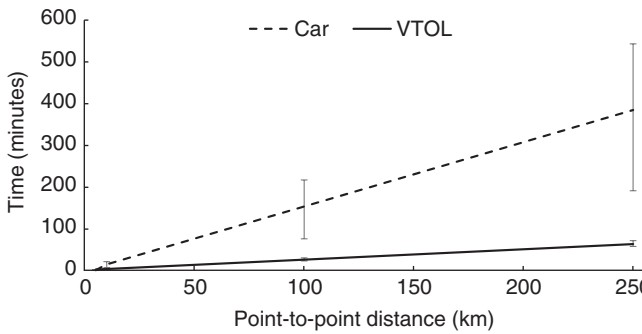

**Fig. 6** Travel-time comparison. Travel-time results for the VTOL and ground-based vehicles (ICEV and BEV) are provided as a function of travel distance. Uncertainty bars show the impact of varying the assumptions for wind speed for VTOLs and urban-highway driving split for cars

using the city average speed and decreases by 50% using the highway average speed.

The travel time for cars is significantly longer than for VTOLs, reflecting their much lower average cruise velocities and, to a lesser extent, more circuitous routing. Travel times as a function of distance for VTOL and ground-based cars are shown in Fig. 6. In the base case (100 km), this equates to an 83% time-saving switching from car to VTOL. This finding is consistent with the estimation of a sixfold travel-time advantage for VTOL travel in Silicon Valley[14]. As seen in Fig. 6, there is no overlap between

uncertainty ranges of both aerial and ground travel times. Even for adverse wind conditions for the VTOL and faster highway travel for the cars, aerial modes still have lower associated journey times than ground-based modes. The time and route certainty for VTOLs, factors that are often unpredictable on the ground, could be valuable for passenger transport.

VTOL throughput will be aided by relatively short travel times and the added degree of freedom associated with aerial mobility. In principle, lines of VTOLs could travel over several vertical layers/stacks. However, regimented operation would likely be enforced by aviation authorities, and for VTOL deployment as an aerial taxi service, physical access to takeoff and landing sites would be limited. Presently, it is hard to gauge how these constraints would compare to limitations of ground-based vehicles. However, it seems plausible that even in a conservative scenario, VTOL throughput would not be a limiting factor to their adoption.

## Discussion

We present the first detailed sustainability assessment of VTOL flying cars. Although VTOLs are faced with economic, regulatory, and safety challenges, we determine that they may have a niche role in a sustainable mobility system. From the results of our assessment, four key insights for VTOL development can be drawn.

First, due to the significantly higher burdens associated with fewer passengers on-board, operators would have to ensure VTOLs fly at near-full capacities for them to outperform

conventional ground-based vehicles. This might be a plausible scenario for two reasons. Current airline service providers already operate with similarly high utilization targets. Also, given the significant time savings of VTOLs over cars, passengers may be motivated to share rides with others to reduce higher costs expected of VTOL trips. While ridesharing in ground-based cars, passengers often have to tradeoff cost for travel time. This is not expected to be the case with VTOLs, with time-saving benefits being potentially important for their adoption. It should be noted here that single-occupant ground-based vehicles also have negative sustainability implications compared with fully loaded cars that combine passenger trips.

Second, VTOLs emit fewer GHGs on a VKT basis compared with ICEVs for trips beyond 35 km. However, the average ground-based vehicle commute is only about 17 km long, with trips exceeding 35 km accounting for under 15% of all vehicle trips[8]. Hence, the trips where VTOLs are more sustainable than ICEVs only make up a small fraction of total annual vehicle-miles traveled on the ground. Subsequently, VTOLs will be limited in their contribution (and role) in a sustainable mobility system. For shorter distances, energy-intensive hover dominates the flight profile, thereby preventing the VTOL from leveraging efficient aerodynamic performance in cruise. VTOL sustainability performance is more advantageous when competing with ground-based vehicles traveling congested routes or indirect routes with higher circuity factors. The comparative energy, emissions, and time-saving benefits of VTOLs are maximized in areas with high congestion or with geographical barriers, which dictate indirect routing for ground-based transport. There could also be an opportunity to displace a portion of short-range regional jet travel with electric VTOLs to reduce GHG emissions. Small jets such as the Embraer 145 with a capacity of 49 passengers have a use-phase well-to-wing GHG burden of 0.10 and 0.20 kg-$CO_2$e PKT$^{-1}$ with load factors of 100% and 50%, respectively[15]. This is comparable to VTOLs with one to three passengers emitting 0.15–0.06 kg-$CO_2$e PKT$^{-1}$ for a 250 km trip.

Third, the GHG emissions of electric VTOLs scale with the carbon intensity of the electricity grid. The carbon intensity of most electric grids are expected to be substantially lower in the future, as more renewable generation is brought on-line. Hence, the benefits of electric VTOLs over conventional fossil-fuel-powered road transportation are expected to grow in the future.

Fourth, lower VTOL emissions, enabled largely by DEP, are not strongly contingent on advances in energy storage. Although superior battery chemistries (and higher specific energies) favor VTOLs' performance over BEVs (owing to greater weight dependencies for the former), they affect range more than they do sustainability impacts for either transport mode, as described in the Methods section.

Related work by Uber[7] and Ullman et al.[6] support the key takeaways from our study. Uber[7] estimates a VTOL energy intensity of about 0.48 kWh km$^{-1}$ at 241 kph for an 80 km trip. No detailed breakdown of the VTOL energy modeling is provided. Under these conditions and assuming four occupants, the VTOL operational energy intensity yielded by our model is 0.43 kWh km$^{-1}$ (about 10% lower). Further, the Ullman et al.[6] model of VTOL range and energy consumption was reproduced according to the physics-based relationships stated in the study. Using Ullman et al.[6] model and input assumptions, the VTOL operational energy intensity is 0.57 kWh km$^{-1}$ for a 1360 kg VTOL at 241 kph for a 100 km trip. Using our baseline input assumptions instead, Ullman's model produces an energy intensity of 0.37 kWh km$^{-1}$ (about 35% lower), which is the same as the output from our model for a fully loaded VTOL. Despite these studies reporting higher energy intensities, the comparison with ground-based vehicles still remains promising. Using the 2020 US

average grid mix, Uber's 0.48 kWh km$^{-1}$ and Ullman's 0.57 kWh km$^{-1}$ translate to GHG emissions of 0.09 and 0.10 kg-$CO_2$e PKT$^{-1}$, respectively (for three passengers on a 100 km trip). These compare favorably to the 0.07 and 0.13 kg-$CO_2$e PKT$^{-1}$ results for the BEV and ICEV, respectively.

Our analysis provides an important first basis for assessing and guiding use-phase VTOL sustainability. Given the dynamic nature of rapid developments in the flying car space, VTOL deployment could emerge differently from our defined base case. This may alter our findings in unpredictable ways. Further, future work should consider the total vehicle cycle burdens for these aircraft, once there is more clarity on material selection, manufacturing processes, design, and disposal. Finally, despite certain sustainability benefits of VTOLs, their feasibility as a future transportation option depends on advances beyond those of a technical nature, including regulation, consumer, and societal acceptance of aerial transport in urban areas.

## Methods

**VTOL key parameter definitions.** The key input parameters used throughout our VTOL physics-based model are defined in Table 1. The table includes the base-case value and corresponding source, as well as the bounds used in the sensitivity analysis for each parameter.

**VTOL range model.** For modeling VTOL range, we begin with the potential energy ($E$) needed to lift the VTOL to a given altitude ($h$). This is considered alongside the system efficiency ($\eta$) used to convert energy stored in the battery to run the electric motor and finally create propulsion through the propellers. For a VTOL with a given takeoff mass ($m$) and given gravity constant ($g$), we have:

$$E = \frac{mgh}{\eta} \qquad (1)$$

The aerodynamics of the VTOL frame convert this potential energy into a distance traveled ($R$), akin to an unpowered glide. The aerodynamic efficiency of the VTOL is specified by the $L/D$ ratio and corresponds to how effectively a VTOL can convert altitude to distance traveled. As such:

$$\frac{L}{D} = \frac{R}{h} \qquad (2)$$

Combining Eqs. (1) and (2), we arrive at the general Eq. (3) for VTOL electric aircraft range:

$$R = E \frac{L}{D} \eta \frac{1}{mg} \qquad (3)$$

Refining this equation to investigate the key performance drivers, we factor the battery mass ($m_b$) into both numerator and denominator terms. This yields the standard Breguet range equation adapted for electric aircraft in Eq. (4)[16]. Overall, the model converts the potential energy required to lift a VTOL to an altitude into range ($R$) achieved from gliding on the aerodynamic wings.

$$R = E \frac{L}{D} \eta \frac{1}{mg} \frac{m_b}{m_b} = E^* \beta \frac{L}{D} \eta \frac{1}{g} \qquad (4)$$

Eq. (4) illustrates key VTOL performance drivers, namely battery technology and design efficiency, as shown in Table 1. Battery-specific energy ($E^*$) is the limiting factor for VTOL range. Lithium-sulfur batteries are being discussed for aerospace applications[17] and are currently being built with pack-specific energies of 400 Wh kg$^{-1}$ [18,19]. The performance characteristics chosen for VTOL batteries (400 Wh kg$^{-1}$ and 1 kW kg$^{-1}$) appear to be plausible in the near future. Several battery chemistries with a practical specific energy upwards of 400 Wh kg$^{-1}$ have been reported[20]. Research and development is underway to improve battery cycle life and specific power. One study reported a cell-specific power of 10 kW kg$^{-1}$ for a certain lithium-sulfur chemistry[21]. Moreover, VTOLs would likely not be regulated by safety requirements around battery packaging as stringent as for BEVs, such as those defined by the Federal Motor Vehicle Safety Standards[22]. Ground-based vehicles are prone to crashes and operational battery wear and tear, thereby warranting such constraints. For VTOLs, reduced overhead packaging weight enables greater realization of cell-specific to battery-specific energy compared with ground-based cars. Further, DEP enables alternative battery topographies for VTOLs, allowing for unique designs of battery warehousing. Another reason VTOLs can adopt advanced batteries earlier is that their service providers would be more likely to pay a premium compared with automotive manufacturers (from a cost-recovery and customer-base willingness to pay standpoint). The alignment of these factors indicates sufficient basis for our assumed value.

Battery mass fraction ($\beta$) describes the mass of the battery packs that can be supported on the airframe. Other important considerations for battery performance include depth of discharge and reserve capacity. Battery health and

cycle life are considered by restricting the usable battery capacity at 80%[3,23]. Current aviation regulations for on-demand, small commuter aircraft mandate 30 min of additional cruise fuel[24]. These regulations are designed for diverting to an alternate airport at the end of long-haul trips and not for short commuter hops with VTOLs, which do not need runway access. Conversely, an aggressive projection specifies 6 min additional cruise, which translates to a narrow safety margin[3]. Our model considers an intermediate reserve battery of 20% capacity for emergencies, which amounts to 15 min of additional cruise time or 5 min of reserve hover time[25]. Incorporating both considerations gives an available battery capacity for standard flight of 60%.

VTOL design efficiency can be broken into aerodynamic efficiency, characterized by the $L/D$ ratio and system efficiency ($\eta$). $L/D$ is a measure of the efficiency of converting potential energy from altitude into distance traveled. $\eta$ is composed of powertrain efficiency (0.9) and propulsive efficiency (0.85 for climb and cruise, and 0.7 for hover)[6,26].

**VTOL energy and GHG emissions modeling**. Diversity of VTOL designs calls for a physics-based approach to primary energy and GHG emission modeling. This section provides the details of the physics-based model and sample calculations for the base-case input values defined in Table 1.

Performance drivers from Eq. (1) directly affect the maximum VTOL range and indirectly affect the emissions results through added battery capacity and mass. To construct a model specifically for energy and emissions, we consider the simplified VTOL flight profile shown in Fig. 1. The VTOL energy model combines climb and descent with cruise due to the uncertainty in the speed profile and transition to/from winged flight during these phases. We selected a cruise altitude of 305 m (1000 ft) for consistency with the Uber[3] report and to meet the minimum safe altitude threshold in Federal Aviation Regulation Part 91.119[27]. As the VTOL is assumed to reach the same altitude during each flight, for simplicity, only the cruise horizontal slant range is assumed to change between different trips. Also note that although $R_{\text{hover}}$ includes hover for both takeoff and landing, the corresponding ground roll is zero. Therefore, given a selected trip length ($R$), the range of each flight phase can be simplified as specified in Eq. (5).

$$R = R_{\text{hover}} + R_{\text{climb}} + R_{\text{cruise}} + R_{\text{descent}} \cong R_{\text{cruise}} \quad (5)$$

With the horizontal slant range known for each phase of flight, we then assess the resulting power requirements. Each mode of flight has a constant average power draw ($P$) over its corresponding time of flight ($t$), which is used to find the overall energy requirements ($E$), as shown in Eq. (6).

$$E = P_{\text{hover}}t_{\text{hover}} + P_{\text{cruise}}t_{\text{cruise}} \quad (6)$$

Power draw is equal to the product of force and velocity in the direction of flight. This velocity is specified as true airspeed (TAS) or the velocity of the aircraft relative to the air. Headwinds or tailwinds do not change the TAS but they do affect the groundspeed (GS). Time of flight and horizontal slant range during each phase is calculated from the GS. Although energy and emissions will change with fluctuating winds and resulting GS for single flights, it is important to note that frequent back and forth along a given air-taxi route would likely average out these changes.

The energy requirement calculated in Eq. (6) is adjusted for a 90% battery charge–discharge efficiency before applying a primary-to-delivered electricity factor for arriving at primary energy[28]. To determine GHG emissions, the computed energy is combined with 2020 US average grid mix projections from the 2017 GREET model, described in the modeling of ground-based vehicles[29]. Regional variations in generation portfolios are captured in the sensitivity analysis, with electric grids from California and Central-and-Southern Plains representing the two bookends for emission factors.

Additional auxiliary power draws from systems such as advanced avionics or passenger comforts (phone charging, heating/cooling, radio, etc.) are excluded, as they would likely have a minor impact on the overall results. For context, an advanced transponder has a power draw of the order of 200 W, which is three orders of magnitude smaller than the power requirements for our VTOL flight[30].

The detailed power and energy calculations for each phase of flight are described below. First, we examine the taxi phase. Power requirements for non-flight activities are aggregated in this segment. This includes wheel-driven taxi to a landing pad from a charging space and vice versa, system power during passenger ingress and egress, and other small draws. As expected time for taxi would be relatively short, about 1 min[3], the energy expended relative to total flight energy is small and hence not accounted for in our model.

Second, hover is examined. Hover is the most energetically intensive phase of the flight profile, because unlike helicopters, blown wing VTOL designs are optimized for cruise. Hover power ($P_{\text{hover}}$) is modeled in Eq. (7) based on momentum theory[31]. For a mean sea level air density ($\rho$), disk loading ($\delta$), and hover system efficiency ($\eta_{\text{h}}$), we have:

$$P_{\text{hover}} = \frac{mg}{\eta_{\text{h}}}\sqrt{\frac{\delta}{2\rho}} \quad (7)$$

Hover power is primarily dependent on rotor disk loading, defined as the VTOL total weight divided by the lifting surface area. The disk loading parameter is

chosen based on data provided in Stoll[32], resulting in a $\delta$ value of 450 N m$^{-2}$ for the VTOL.

In addition to disk loading ($\delta$), we use a hover system efficiency ($\eta_{\text{h}}$) of 0.63, which incorporates a powertrain efficiency of 0.9 and propulsive efficiency of 0.7 (instead of 0.85) to account for lifting inefficiencies[6]. Hover relates to vertical takeoff and landing, as well as intermediate loitering, and thus has no associated ground roll[6].

Eq. (7) culminates in an average power requirement of 250.6 kW:

$$P_{\text{hover}} = \frac{1187.5\ \text{kg} * 9.81\ \text{m s}^{-2}}{0.63}\sqrt{\frac{450\ \text{N m}^{-2}}{2 * 1.22\ \text{kg m}^{-3}}} = 250.6\ \text{kW}$$

For a minute-long hover (2-legs of 30 s each), total primary energy required for this leg of the flight (constant across each trip) is 40.9 MJ (accounting for charge–discharge and primary-to-delivered energy efficiencies of 90% and 40.8%, respectively):

$$\text{Primary energy for hover} = \frac{250.6\ \text{kW} * 60\ \text{s}}{1000 * 0.408 * 0.9} = 40.9\ \text{MJ}$$

Third, we model climb and descent, which are modeled in the same way as cruise for three main reasons. First, the energy required in excess of cruise performance to climb and accelerate is approximately balanced out by the lower energy required during the descent and deceleration segment, such that assuming cruise performance for the whole duration is a good approximation. Second, limited data are available indicating how the VTOL TAS and corresponding $L/D$ would change throughout climb and descent. Finally, due to the cruise altitude of 1000 ft and the assumed rate of climb (ROC) and rate of descent (ROD) of 1000 fpm, the climb and descent phases have a duration of only 2 min, which is only a small portion of the 25 min flight in the base case.

However, if/when an accurate velocity profile and VTOL configuration is made available, a higher fidelity modeling approach may be used. In this case, climb would be modeled separately from cruise and power requirements would be split up into two distinct parts. First, the potential energy used to lift the VTOL to a given altitude is converted to power ($P_{\text{climb,PE}}$) by dividing by the time for climb ($t_{\text{climb}}$). This duration is found by specifying the ROC and the target altitude ($h$).

$$P_{\text{climb,PE}} = \frac{mgh}{t_{\text{climb}}} \quad (8)$$

Next, we consider the power necessary to overcome aerodynamic forces during climb ($P_{\text{climb,D}}$). A flight path angle ($\gamma$) is specified for the VTOL during climb. Thus, we calculate the climb TAS ($V_{\text{climb}}$) using $\gamma$ and ROC. $L/D_{\text{climb}}$ and $V_{\text{climb}}$ enable determination of the power necessary to overcome aerodynamic forces during climb.

$$P_{\text{climb,D}} = \frac{mg}{L/D_{\text{climb}}}V_{\text{climb}} \quad (9)$$

Combining these two power elements in Eqs. (8) and (9) yields Eq. (10), which also incorporates climb system efficiency ($\eta_{\text{c}}$):

$$P_{\text{climb}} = \left(\frac{mh}{t_{\text{climb}}} + \frac{m}{L/D_{\text{climb}}}V_{\text{climb}}\right)\frac{g}{\eta_{\text{c}}} \quad (10)$$

Now, we reduce the cruise altitude over time of flight in climb term to be equivalent to the ROC. This yields the final power Eq. (11) for climb ($P_{\text{climb}}$), which would have to be integrated over the velocity and $L/D$ profile during the climb phase.

$$P_{\text{climb}} = \frac{mg}{\eta_{\text{c}}}\left(\text{ROC} + \frac{V_{\text{climb}}}{L/D_{\text{climb}}}\right) \quad (11)$$

Alternatively, we arrive at the same modeling equation using a free body diagram of a VTOL in climb, in which we observe the four forces acting on an aircraft: Lift (**L**), Weight (**W**), Thrust (**T**), and Drag (**D**). VTOL weight is a product of its takeoff mass ($m$) and acceleration due to gravity ($g$). A diagram of the relationship is provided in Supplementary Fig. 5.

The power needed for climb can be simplified as the product of thrust produced by the VTOL and the TAS. From the momentum conservation principle, the thrust force is approximately resolved into the opposing drag force and a small component of the VTOL weight, as shown in Supplementary Fig. 5. Owing to winged VTOL design, the component of induced velocity for cruise flight, if any, is neglected. Next, the ($V_{\text{climb}}\sin\gamma$) term is expressed as ROC. Using a small flight path angle assumption, we consider weight to be approximately equal to lift. Drag force can be found through dividing weight by $L/D_{\text{climb}}$.

$$P_{\text{climb}} = \frac{TV_{\text{climb}}}{\eta_{\text{c}}} = \frac{V_{\text{climb}}}{\eta_{\text{c}}}(mg\sin\gamma + D) = \frac{g}{\eta_{\text{c}}}\left(mV_{\text{climb}}\sin\gamma + \frac{mV_{\text{climb}}}{L/D_{\text{climb}}}\right) = \frac{mg}{\eta_{\text{c}}}\left(\text{ROC} + \frac{V_{\text{climb}}}{L/D_{\text{climb}}}\right)$$

$$(12)$$

Fourth, cruise flight is modeled using a simple force balance, depicted through the free body diagram shown in Supplementary Fig. 6. We assume equal force couples in steady, non-accelerated flight (equal lift and weight, and equal thrust and drag). We then use the thrust and cruise TAS ($V$) of the VTOL to find the

power draw during cruise ($P_{cruise}$). As assumed, thrust produced is equal to the drag force and is found by dividing VTOL weight ($W$) by $L/D$. This yields Eq. (13), which also considers cruise system efficiency ($\eta_c$).

$$P_{cruise} = \frac{mg}{\frac{L}{D}} \frac{V}{\eta_c} = \frac{(1187.5 \text{ kg} * 9.81 \text{ m s}^2 * 66.7 \text{ m s}^{-1})}{17 * 0.765 * 1000} = 59.7 \text{ kW} \quad (13)$$

A cruise power ($P_{cruise}$) of ~59.7 kW is calculated. For determining primary energy for cruise, we first use Eq. (5). In the base-case scenario:

$$R_{cruise} \cong 100 \text{ km}$$

Using our 150 mph cruise velocity, a corresponding base-case cruise time of about 24.9 min is obtained. Finally,

$$\text{Primary energy for cruise} = \frac{59.7 \text{ kW} * 24.9 \text{ min} * 60 \text{ s min}^{-1}}{1000 * 0.408 * 0.9} = 243.0 \text{ MJ}$$

Adding the individual primary energy associated with each leg of the flight profile gives a total base-case VTOL primary energy use of about 284 MJ. The primary energy is converted to GHG emissions by multiplying by 0.408 to convert back to delivered electricity, then multiplying by 0.135 kg-$CO_2$e $MJ^{-1}$ to get 15.7 kg-$CO_2$e.

Finally, we incorporate reserves. Present reserve requirements for VTOLs remain unclear without official regulation from aviation authorities. Existing FAA regulations for Part 135 aircraft mandate 30 min of additional cruise fuel. This regulation is designed for diversion to alternate airports at the end of long-haul trips, not for shorter commute hops with VTOLs that do not need a runway to land. Therefore, we use the current FAA regulation as a conservative upper bound for safety.

At the lower end are two predictions of 6 and 10 min reserves for cruise[6,7]. Although these seem more appropriate to the functionality of VTOLs, safety would be of the highest concern for consumer adoption. Emergency scenarios or potentially adverse conditions call for greater robustness. Our model uses an intermediate value of 20% of total battery capacity as reserve[25], amounting for 15 min of additional cruise time, or 5 min of reserve hover time.

**Ground-based vehicles energy and GHG emissions modeling**. Ground-based passenger vehicles are modeled as generic mid-sized, light-duty ICEVs and BEVs with fuel economy, powertrain efficiency, and battery-specific energy projected for the year 2020[10]. Although battery-specific energy determines range for BEVs (as for VTOLs), it has minor impacts on fuel economy (which relates more to electric motor efficiency). For functional equivalence with our VTOL, a long-range (340 km, 210 mi) BEV is chosen. This range accounts for battery health.

Four factors need to be considered to assess the energy and emissions for cars. First, the fuel economy. Fuel economy values for urban and highway driving are adjusted for on-road (real-world) conditions using the five-cycle testing method (Supplementary Table 1)[11]. The five-cycle test is representative of typical US commuting, in that it covers five distinct driving patterns (including aggressive driving, extreme ambient temperatures, and heating and air-conditioning usage), and considers an equivalent payload weight. Scenarios corresponding to the bookend fuel economies of urban and highway driving are modeled as extremes and described as sensitivities. For obtaining a baseline value, we compute the combined fuel economy as a harmonically weighted average of 55% city/45% highway driving activity, as specified by the US EPA[11]. For details, see Supplementary Eq. (1), as part of Supplementary Note 1.

Second, the added weight from incremental payload and components, which increase its fuel consumption. Payload-induced fuel consumption increase values of 0.073 and 0.27 L equivalent per 100 km per 100 kg, based on EPA five-cycle testing, were used for the BEV and ICEV, respectively[33]. An equivalency factor of roughly 8.9 kWh $L^{-1}$ was applied.

Third, the fuel carbon intensity, which dictates the emissions profile of the vehicle. The 2017 GREET model was used as a basis for these values[29]. Similar to the electric VTOL, the BEV GHG emissions are driven by the charging grid. For the base case, we assumed the 2020 US average distributed mix from GREET (34% coal, 28% natural gas, 19% nuclear, 8% hydro, 8% wind, and 3% other), with the delivered electricity corresponding to a GHG-100 intensity of 0.135 kg-$CO_2$e $MJ^{-1}$. The corresponding efficiency for primary-to-delivered grid electricity is 40.8%. This factor is modified from GREET, which accounts for upstream energy impacts of nuclear-based grid electricity using the NREL US LCI database[34] (see Supplementary Note 2 for details). We assume a 90% battery charge/discharge efficiency for the BEV, consistent with the VTOL[28]. The ICEV platform is powered by conventional E10 gasoline, with a lower heating value of 119.6 MJ $gal^{-1}$, well-to-wheel primary energy use of 1.28 MJ per MJ of delivered fuel energy, and a well-to-wheel carbon intensity of 0.093 kg-$CO_2$e per MJ of delivered energy.

Fourth, circuitous (indirect) routing is an important consideration specific to ground-based modes. Ground-based routes are typically longer than the shortest distance between two points. This circuity factor is highly variable and depends on geographic location, urban density, preferred routing, and road-network connectivity. We used the US average circuity factor of 1.20 to determine the effective performance of ground transport modes[5]. For context, in an assessment of average circuity factors by country, Belarus has the most direct routing (average circuity factor 1.12), whereas Egypt has the least direct routing (average circuity

factor 2.10)[5]. Dividing the BEV range of 320 km by the US average circuity factor provides a point-to-point BEV range of ~280 km.

**Travel-time modeling**. The travel time for VTOL flight was modeled based on the simple flight profile shown in Fig. 1. This begins with an assumed 30 s hover for takeoff. We then assume a ROC of 1000 fpm to 1000 ft cruise altitude, with a similar ROD, followed by a 30 s hover to landing[35]. Cruise TAS of 150 mph follows leading VTOL design[32]. The resulting GS and travel time are calculated, with potential headwinds and tailwinds of 30 knots each modeled for their sensitivities.

For ground-based vehicles, travel time is inherently more uncertain, changing with chosen route, traffic conditions, time of day, region, and weather[36]. Here, a simple model incorporating effective distance and average velocity is employed for estimating commute time. For a general estimate of travel time, we first derive velocities for city and highway driving individually. This is done by computing the weighted average of factors specified in testing guidelines (Supplementary Table 2)[11]. The average speeds obtained are used as the basis for modeling travel times corresponding to all-city and all-highway driving patterns. For a baseline average speed—consistent with our fuel economy modeling approach—we use a harmonically weighted average of 55% city/45% highway driving speeds[11]. Equations for modeling travel time, consistent with the five-cycle testing procedure, are provided in Supplementary Note 3 (see Supplementary Eqs. (2), (3), and (4)).

## Data availability
The authors declare that all data supporting the findings of this study are available within the paper and its Supplementary Information files.

## Code availability
Supporting code is available from the authors upon reasonable request.

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

## Acknowledgements

We thank James E. Anderson and Sandy L. Winkler (Ford Motor Company), William J. Fredericks (Advanced Aircraft Company), and Geoffrey M. Lewis, Carlos E.S. Cesnik, and Nilton O. Renno (University of Michigan) for helpful discussions. This study was supported by Ford Motor Company through their Summer Internship Program and a Ford-University of Michigan Alliance Project Award (No. N022546-00).

## Author contributions

T.J.W., R.D.K., H.C.K., and G.A.K. designed and supervised the research. J.R.M. provided the governing VTOL model and parameters. A.K., N.J.F., and J.H.G. performed the research and wrote the paper, with inputs from all co-authors.

## Additional information

**Competing interests:** The authors declare no competing interests.

