## [Peer Review File · Nature Communications]

Reviewers' comments:

Reviewer #1 (Remarks to the Author):

This research assesses the energy use and greenhouse gas emissions of “flying cars,” which are defined as 4-5 seat, battery-powered, vertical take-off and landing (VTOL) aircraft. The results are compared to equivalent trips by gasoline and electric ground vehicles. The authors conclude that VTOLs have lower emissions than conventional vehicles and higher emissions than electric vehicles under most conditions examined.

This paper is generally clear and well-executed and covers a current topic relevant to energy and environmental policy. The primary weakness of the paper is the narrow framing of the VTOL use case and the limited sensitivity analysis, which call into question the generality of the conclusions. In specific, the authors conclude that “VTOLs could play a niche role in a sustainable mobility system,” which, to be fair, is not a strong claim, but it still isn’t well supported by the results.

Roughly speaking, gasoline vehicles have no role whatsoever in a sustainable mobility system, so the fact that VTOLs outperform them is no claim to sustainability. The fact that ground-based EV’s outperform VTOLs in virtually every case examined undermines their usefulness with respect to sustainability.

However, there are some factors that the authors didn’t consider that could make a better case for VTOLs. First, the passenger loading. Passenger cars rarely carry 4-5 passengers at once. Are VTOLs more likely to fly full? Passenger loading has a dramatic effect on GHG emissions and energy use per person-trip, and could very well dominate the comparison between VTOLs and BEVs. Second, regional jets are very inefficient compared to BEVs. Could VTOLs displace some commuter jet traffic? What would the impact be? I think a (niche) sustainability claim could be better made if VTOLs can somehow displace fossil-fueled aircraft or single-passenger cars. More specific comments follow.

p. 2 L. 38: please briefly describe DEP.

p. 4 L.84: provide a reference for the Breguet equation

Eqn. 1: You should define all the parameters immediately before or after an equation. I know they appear in Table 1, but still.

p. 5 L. 102: How does 20% capacity correspond to minutes?

Figure 1. What’s the basis for the choice of cruise height?

Figure 2. This might be better presented as a wedge graph.

p. 10 L. 189: These L/D values seem pretty high compared to small conventional aircraft. Should you have a lower low-end value or is there a good basis for 15?

Sensitivity analysis: Sensitivity results should be summarized in a figure or table. The carbon intensity of electricity varies regionally and this should be discussed or included in the analysis.

p. 11 L. 211: Why can't Li-S battery advances also be applied to ground vehicles? Why is housing not as important for VTOLs?

Methods: The calculations that comprise most of the SI should be moved to Methods.

Reviewer #2 (Remarks to the Author):

1. The discussion of a high-aspect ratio blown wings enabled by DEP on lines 40 and 315 references a study of CTOL aircraft, but the conclusions of this study are not applicable to VTOL aircraft. Specifically, the cited study discusses retaining CTOL performance (e.g., a suitably low stall speed) while reducing wing area to a size better suited to efficient cruise flight; however, VTOL aircraft are, in general, not constrained to such CTOL performance considerations.

2. The key parameters for VTOL aircraft in Table 1 include a cruise speed of 200 mph and a cruise L/D of 17. However, while the cited Uber white paper does assume an L/D of 17, this is at 150 mph, and an L/D of 13 at 200 mph is assumed.

3. No rationale is given for the choice of the assumed climb L/D, which is much lower than the cruise L/D. Typically, such aircraft can reasonably be expected to climb and descend at close to their maximum L/D.

4. The model and parameters given result in the climb and descent phases covering about 5 km of ground distance. It appears that this complete distance is used even when the entire flight distance is less than 5 km, resulting in disproportionately high energy usage for such shorter flights. No rationale is given for this decision. It is certainly conceivable that such an aircraft would begin its descent at a lower altitude than the stated cruise altitude, if the total flight distance is too short to reach the stated cruise altitude.

5. No rationale is given for the decision to climb and descend at the flight path angle given by arctangent of the reciprocal of L/D_{climb} . Such aircraft may be more likely to climb and descend at or near their cruise airspeed, to shorten the travel times.

Reviewer #3 (Remarks to the Author):

For the reader who is new to VTOL you need to differentiate them from rotorcraft also being considered as "flying cars". Pictures of VTOLs and rotorcraft may help here.

Line 61: Need to spell out ICEVs and BEVs on first use for the reader new to the field.

Figure 1 in the manuscript is messed up. S1 is correct.

What is "S1" and figures "S5", "S6" etc? I figured out that they referred to the supplemental material. I guess I am not familiar with Nature Communications' handling of papers and support material. Should the reader be told this or will the links be active in the final paper?

Figures in your sensitivity analysis section would be helpful for the reader.

Note that I have not checked the equations. I assume the authors have compared their result to those of German and Ullman for verification. They need to state this or other support for why I should believe their results.

Why no uncertainty bars on the BEVs in Fig S6?

Be nice to end with some conclusions (restated from the abstract) and next steps. The paper as it is now, just stops.

We would like to thank the editor and the reviewers for their valuable feedback and comments. Based on reviewer comments, we have made changes to the manuscript that enhance its overall quality. Please find our responses to the comments below.

Reviewer 1

Comment 1: The primary weakness of the paper is the narrow framing of the VTOL use case and the limited sensitivity analysis, which call into question the generality of the conclusions. In specific, the authors conclude that “VTOLs could play a niche role in a sustainable mobility system,” which, to be fair, is not a strong claim, but it still isn’t well supported by the results. Roughly speaking, gasoline vehicles have no role whatsoever in a sustainable mobility system, so the fact that VTOLs outperform them is no claim to sustainability. The fact that ground-based EV’s outperform VTOLs in virtually every case examined undermines their usefulness with respect to sustainability.

Response

We agree that a broader and more robust claim to VTOL sustainability can be made by exploring differences in passenger loading relative to ground-based cars (aligned with Comment 2), and adding more scenarios to the sensitivity analysis. Since the first-wave of VTOLs would likely operate as aerial-taxis, service providers would target near-full occupancy (roughly 3 passengers or 262.5 kg¹, in addition to the pilot) from a revenue and utilization-maximization standpoint. This is similar to current commercial airlines. Passengers could be incentivized to share VTOL rides given the significant time savings from flying. In contrast, most U.S. sedans have a relatively small occupancy of 1.54 passengers per vehicle on average.² Given this expected occupancy difference, we explore a scenario in the sensitivity analysis where we normalize burdens per PKT when comparing VTOLs to cars. Reporting results per PKT is a common practice in life cycle studies, especially when comparing two different transport modes.³ When incorporating differences in expected occupancies, we find that VTOLs outperform *both* ICEVs and BEVs in the base-case (roughly 52% and 8% lower emissions). Further, we determine that in such a scenario, breakeven VTOL emissions relative to BEVs occur at about 60 km.

We highlight that the benefits from saved commute times (83% in the 100 km base-case) could enable greater ride-sharing (and subsequently higher load factors) for VTOLs relative

to ground-based cars. Further, we supplement these findings with a more detailed sensitivity analysis for both aerial and ground-based modes, including varying passenger loading, battery specific energy, grid carbon-intensity, wind, L/D ratio, and powertrain efficiency to test the impact of our assumptions. In doing so, we are now able to assess VTOL sustainability more robustly. The major takeaway from the sensitivity analysis, however, remains that VTOLs only contribute to sustainability if used as shared-service with high occupancies.

On a vehicle-level basis (single occupant), VTOL emissions per VKT drop below ICEVs beyond 35 km. The average ground-based vehicle trip is only about 17 km long, with trips over 35 km accounting for under 15% of all vehicle trips.² Hence, the trips where VTOLs are more sustainable than conventional passenger-cars only make up a relatively smaller fraction of total annual vehicle-miles traveled on the ground. Subsequently, VTOLs might be limited in their contribution (and role) in a sustainable mobility system. These points have been elaborated upon in the Discussion section.

New content also now includes references to the following studies:

- 1) Federal Aviation Administration (FAA). Advisory Circular: Aircraft Weight and Balance Control. https://www.faa.gov/documentLibrary/media/Advisory_Circular/AC120-27E.pdf (2005).
- 2) US DOT. National Household Travel Survey, 2017. <https://nhts.ornl.gov/>. Accessed December 11, 2018.
- 3) Chester, M., & Horvath, A. High-speed rail with emerging automobiles and aircraft can reduce environmental impacts in California's future. *Environmental research letters*, 7(3), 034012., 2012.

Change to Manuscript

Please see Figures 4, 5a, and 5b in the revised manuscript for a summary of updated findings. Text has also been updated throughout the manuscript, as described in the response to this comment.

Comment 2: First, the passenger loading. Passenger cars rarely carry 4-5 passengers at once. Are VTOLs more likely to fly full? Passenger loading has a dramatic effect on GHG emissions and energy use per person-trip, and could very well dominate the comparison between VTOLs and BEVs.

Response

We agree that it would be more realistic to add a scenario in the sensitivity analysis considering differences in expected average occupancies for both modes. While the BEV has 1.54 passengers on average (with the driver considered a passenger), a VTOL could have 3 passengers in addition to the pilot. This narrative is introduced after results are reported on a like-for-like vehicle-level (piloted VTOL/single-occupant sedan) basis. As indicated in Comment 1, the first-wave of VTOLs would likely operate as aerial-taxis, and service

providers would target high utilization for maximizing revenues, similar to current commercial airlines. Passengers could be incentivized to share VTOL rides given the significant time savings from flying. In contrast, most U.S. sedans have a relatively small occupancy of 1.54 passengers per vehicle. Hence, burdens are normalized per PKT and impacts of variable passenger loading for each mode are explored in the Sensitivity Analysis. We find that fully-loaded 3 passenger piloted VTOLs have 8% lower emissions-intensity than an average 1.54 passenger BEV. The primary takeaway in the Discussion section is centered around the need for VTOLs to be operated as a high-occupancy rideshare service, since they are only found to be more sustainable with greater passenger loading. Our analysis and subsequent findings have been updated for the same. Please see the in-text addition to the manuscript below.

Change to Manuscript

Original paragraph beginning line 69:

“Our analysis considers transporting 400 kg (equivalent to 4-5 passengers), over point-to-point distances from 1 to 250 km, either by flying or ground-based vehicles. The base-case is assessed for a 100 km distance, the approximate span of a major city.”

New paragraph with updated comparison beginning line 59:

“Our analysis first focuses on a vehicle-to-vehicle comparison with one occupant (i.e. the pilot or driver) traveling point-to-point distances ranging from 5 to 250 km. The base-case is assessed for a 100 km distance. As part of a sensitivity analysis, we compare the results on a passenger-kilometer traveled (PKT) basis. The VTOL is assumed to have 3 passengers and one pilot (i.e. 4 occupants) since it will most likely be used in a transportation-as-a-service business model where service providers seek to maximize utilization rates. Ground-based cars are assumed to be personally owned with a typical loading of 1.54 passengers/occupants.¹ Modeling details are available in the Methods section, with uncertainties explored in the Sensitivity Analysis.”

New paragraph with updated comparison beginning line 144:

“VTOLs are initially likely to operate as aerial-taxis and service providers would target near-full occupancy from a utilization-maximization standpoint, similar to current commercial airlines. Passengers could be incentivized to share VTOL rides given the significant expected time savings from flying. Thus, it seems likely that the average occupancy of VTOLs will be greater than for conventional passenger cars. Given an expected occupancy difference, it can be argued that the emission burdens between VTOLs and ground vehicles should be compared on a per passenger-kilometer traveled (PKT) basis, rather than the per vehicle kilometer traveled (VKT) basis shown in Figure 4. The results of such an assessment are described in the Sensitivity Analysis.”

New content also now includes a reference to the following report:

1) US DOT. National Household Travel Survey, 2017. <https://nhts.ornl.gov/>. Accessed December 11, 2018.

Comment 3: Second, regional jets are very inefficient compared to BEVs. Could VTOLs displace some commuter jet traffic? What would the impact be? I think a (niche) sustainability claim could be better made if VTOLs can somehow displace fossil-fueled aircraft or single-passenger cars.

Response

We agree that there may be an opportunity to displace regional commuter jet traffic with electric VTOLs to reduce GHG emissions. The Discussion section now includes this comparison. Please see the addition to the manuscript below.

Change to Manuscript

New content beginning line 277:

“There could also be an opportunity to displace a portion of short-range regional jet travel with electric VTOLs to reduce GHG emissions. Small jets such as the Embraer 145 with a capacity of 49 passengers have a use-phase life cycle GHG burden of 0.10 and 0.20 kg-CO₂e/PKT with load factors of 100% and 50%, respectively.¹ This is comparable to VTOLs with 1-3 passengers emitting approximately 0.15-0.06 kg-CO₂e/PKT for a 250 km trip.”

New content also now includes a reference to the following paper:

1) Chester, M. V., & Horvath, A. (2009). Environmental assessment of passenger transportation should include infrastructure and supply chains. *Environmental Research Letters*,4(2), 024008. doi:10.1088/1748-9326/4/2/024008

Comment 4: p. 2 L. 38: please briefly describe DEP.

Response

We agree that a more detailed definition of DEP would be helpful here. Please see the addition to the manuscript below.

Change to Manuscript

Original paragraph beginning line 38:

“Electric powertrains are inherently more efficient than combustion powertrains and, as with ground-based transportation, electrification is a likely propulsion strategy for sustainable short-range aerial mobility. Distributed electric propulsion (DEP) is the critical enabler for VTOL performance. In addition to significant noise reduction, DEP enables aerodynamically optimised designs that allow efficient VTOL energy performance. Aerodynamic improvements include articulating propellers and high aspect-ratio blown wings. DEP could facilitate VTOL success in the urban aerial-taxi space, where conventional helicopters or vertical-lift aircraft have struggled.”

New paragraph with definition beginning line 35:

“Electrification is a propulsion strategy for improving the sustainability of both aerial and ground-based transportation-modes, owing to the superior efficiency of electric powertrains compared to combustion engines. One critical efficiency-enabler for VTOLs is distributed electric propulsion (DEP), which utilizes physically smaller, electrically-driven propulsors. Subsequently, propulsors can be used with greater flexibility to leverage the benefits of aero-propulsive coupling and improve performance compared to more traditional designs.¹ This enables aerodynamically optimized designs, such as articulating propellers and high aspect-ratio blown wings, which allow efficient VTOL energy performance and significant noise reduction. DEP could facilitate VTOL success in the urban aerial-taxi space, where conventional helicopters or vertical-lift aircraft have struggled.”

New paragraph also now includes a reference to the following paper:

- 1) Kim, Hyun D., et al. A Review of Distributed Electric Propulsion Concepts for Air Vehicle Technology, in AIAA/IEEE Electric Aircraft Technologies Symposium, 2018. Aerospace Research Central (ARC) <https://doi.org/10.2514/6.2018-4998> (2018).

Comment 5: p. 4 L.84: provide a reference for the Breguet equation.

Response

We agree that a reference to the standard Breguet range equation would be helpful. The method for converting from the standard Breguet range equation to the Breguet range equation for electric aircraft is provided in the methods section. Please see the addition to the manuscript below.

Change to Manuscript

Original Paragraph beginning Line 84:

“Diversity of VTOL designs calls for a physics-based approach to energy modelling. The most optimal designs will utilize the takeoff and landing accessibility of hover, similar to a helicopter, and the cruise efficiency of a winged aircraft. Equation (1) is the standard Breguet range equation adapted for electric aircraft. This basic model converts the potential energy required to lift a VTOL to altitude, into range (R) achieved from gliding on the aerodynamic wings.”

$$R = E^* \beta \frac{L}{D} \eta \frac{1}{g} \quad (1)$$

New Paragraph with reference beginning Line 325:

“In order to model the VTOL range, we begin with the potential energy (E) needed to lift the VTOL to a given altitude (h). This is considered alongside the powertrain efficiency of the system (η) used to convert stored energy in the battery to run the electric motor and finally create propulsion through the propellers. For a VTOL with a given maximum takeoff mass (W) and given gravity constant (g), we have:

$$E = \frac{Wgh}{\eta} \quad (1)$$

The aerodynamics of the VTOL frame convert this potential energy into a distance traveled (R), akin to an unpowered glide. The aerodynamic efficiency of the VTOL is specified by the lift to drag ratio (L/D) and corresponds to how well a VTOL can convert altitude to distance traveled. As such:

$$\frac{L}{D} = \frac{R}{h} \quad (2)$$

Combining Equations (1) and (2), we arrive at the general Equation (3) for VTOL electric aircraft range:

$$R = E \frac{L}{D} \eta \frac{1}{Wg} \quad (3)$$

Refining this equation to investigate the key performance drivers, we factor the battery mass (m_b) into both numerator and denominator terms. This yields the standard Breguet range equation adapted for electric aircraft in equation (4).¹ Overall, the model converts the potential energy required to lift a VTOL to altitude, into range (R) achieved from gliding on the aerodynamic wings.

$$R = E \frac{L}{D} \eta \frac{1}{Wg} \frac{m_b}{m_b} = E \beta \frac{L}{D} \eta \frac{1}{g} \quad (4)$$

New paragraph also now includes a reference to the following paper:

1) Greatrix, David R. Introduction to Atmospheric Flight. Powered Flight, 29–62. Springer https://doi.org/10.1007/978-1-4471-2485-6_2 (2012).

Comment 6: Eqn. 1: You should define all the parameters immediately before or after an equation. I know they appear in Table 1, but still.

Response

We agree that explicitly defining parameters as they appear in an equation would make it easier to follow for readers. For instance, please see the addition to the manuscript below.

Change to Manuscript

Supplementary Paragraph originally appearing in SI:

“We begin with the potential energy needed to lift the VTOL to a given altitude (h). This is considered alongside the powertrain efficiency of the system used to convert stored energy in the battery to run the electric motor and finally create propulsion through the propellers.”

$$E = \frac{Wgh}{\eta} \quad (1)$$

New Paragraph (in manuscript) beginning Line 325:

“In order to model the VTOL range, we begin with the potential energy (E) needed to lift the VTOL to a given altitude (h). This is considered alongside the powertrain efficiency of the system (η) used to convert stored energy in the battery to run the electric motor and finally create propulsion through the propellers. For a VTOL with a given maximum takeoff mass (W) and given gravity constant (g), we have:”

$$E = \frac{Wgh}{\eta} \quad (1)$$

Comment 7: p. 5 L. 102: How does 20% capacity correspond to minutes?

Response

We agree that adding the corresponding time in minutes would be helpful to the reader.

Change to Manuscript

Original Sentence beginning Line 102:

“Our model considers an intermediate reserve battery of 20% capacity for emergencies.”

New Sentence beginning Line 370:

“Our model considers an intermediate reserve battery of 20% capacity for emergencies, which amounts to 15 minutes of additional cruise time, or 5 minutes of reserve hover time.”

Comment 8: Figure 1. What’s the basis for the choice of cruise height?

Response

The cruise height is based on the input data used in the referenced Uber (2018) report. It is a balance between safety and energy consumption. Regarding safety, a high cruise height allows for better clearance of ground-based objects and more time for addressing in-flight issues. Federal Aviation Regulation Part 91.119 lists a minimum safe altitude of 1,000 feet over congested areas, which will be the most common operating environment for VTOLs. Regarding energy consumption, a low cruise height reduces the energy intensive climb phase of the flight plan and allows the VTOL to reach the more energy efficient cruise phase more quickly. Therefore, the minimum safe altitude threshold of 1,000 feet was selected for our modeling since it satisfies the safety requirement while also being more energy efficient.

Change to Manuscript

New sentence added to Methods section beginning on Line 390:

“We selected a cruise altitude of 305 meters (1000 feet) for consistency with the Uber¹ report and to meet the minimum safe altitude threshold in Federal Aviation Regulation Part 91.119.34²”

New sentence also now includes a reference to the following reports:

1) Uber Elevate. EVTOL Vehicle Requirements and Missions.

[https://s3.amazonaws.com/uber-static/elevate/Summary Mission and Requirements.pdf](https://s3.amazonaws.com/uber-static/elevate/Summary+Mission+and+Requirements.pdf) (2018). Accessed June 6, 2018.

2) US GPO. Part 91 – General Operating and Flight Rules (14 C.F.R. § 91.119).

https://www.ecfr.gov/cgi-bin/text-idx?c=ecfr&sid=3efaad1b0a259d4e48f1150a34d1aa77&rgn=div5&view=text&node=14:2.0.1.3.10&idno=14#se14.2.91_1119 (2018). Accessed December 11, 2018.

Comment 9: Figure 2. This might be better presented as a wedge graph.

Response

We agree that this figure would make for a stronger visual being presented as a wedge graph, and have made necessary adjustments.

Change to Manuscript

Please see the new version of Figure 2 (now Figure 3) in the manuscript. Note that it has been color coded to match the phases of flight in Figure 2.

Comment 10: p. 10 L. 189: These L/D values seem pretty high compared to small conventional aircraft. Should you have a lower low-end value or is there a good basis for 15?

Response

We agree a lower low-end value could be used for the L/D. We have lowered it to a value of 13 in the sensitivity analysis in alignment with the referenced Uber (2016) report. Using a L/D of 13 instead of 17 increases the VTOL GHG emissions per VKT by 24%.

Change to Manuscript

Please see the sensitivity analysis presented in Figure 5a.

Comment 11: Sensitivity analysis: Sensitivity results should be summarized in a figure or table. The carbon intensity of electricity varies regionally and this should be discussed or included in the analysis.

Response

We agree that it would be more effective to have the sensitivity analyses shown graphically. Further, we have included regional variations in electric grids as part of our sensitivity analysis, considering emission factors specific to California and the Central-and-Southern Plains grid (from the GREET model) as our bookends. Additions are both in-text, and graphical. The in-text additions are described below.

Change to Manuscript

Original Line 299:

“To determine GHG emissions, the computed energy is combined with 2020 U.S. average grid mix projections from the 2017 GREET model, described in modelling of ground-based vehicles.”

Added sentence beginning Line 411:

“To determine GHG emissions, the computed energy is combined with 2020 U.S. average grid mix projections from the 2017 GREET model, described in modeling of ground-based vehicles.¹ Regional variations in generation portfolios are captured in the sensitivity analysis,

with electric grids from California, and Central-and-Southern Plains representing the two bookends for emission factors.”

New sentence also now includes a reference to the GREET model:

- 1) Argonne National Laboratory (ANL). The Greenhouse Gases, Regulated Emissions, and Energy Use in Transportation (GREET) Model Software: GREET 1 (2017).

Added sentence beginning Line 163:

“First, changing the 2020 electrical grid carbon intensity from the U.S. average mix to the California, and Central-and-Southern Plains grids results in a 52% decrease and 41% increase in emissions, respectively. Note that a similar effect will be seen with the BEV when comparing these VTOL results to the BEV baseline.”

Also see Figure 5a for the graphical representation of the sensitivity analysis results.

Comment 12: p. 11 L. 211: Why can't Li-S battery advances also be applied to ground vehicles? Why is housing not as important for VTOLs?

Response

We now model impacts of increasing battery specific energy (from 250 to 400 Wh/kg) on BEV emissions. We find that increasing specific energy from 250 to 400 Wh/kg reduces base-case BEV emissions by less than 4%. While Li-S batteries may be used in BEVs in the future, in the interests of being functionally equivalent and having similar BEV and VTOL ranges, we model the BEV with batteries of 250 Wh/kg. This value allows us to be consistent with the 2020 projections from the Elgowainy et al. study that we reference for modeling the ground-based cars. Fundamentally, the energetics of a BEV are dictated by the efficiency of the electric motor, and lesser so by the battery specific energy.

Next, the reason VTOLs have flexibility with battery housing owes to the fact that they are likely to not be regulated by stringent FMVSS standards like BEVs (which are more prone to crash accidents and operational wear and tear). Further, DEP allows for unique topologies of battery placements on the VTOL airframe, in design configurations not typical for conventional BEVs. Another reason VTOLs can adopt advanced Li-S batteries earlier is that their service providers would be more likely to pay capital premiums compared to automotive OEMs (from a cost-recovery and customer-base willingness to pay standpoint). Finally, we found that VTOL and BEV emissions are not strongly contingent on or impacted by advances in energy storage, which has been highlighted in the Discussion section.

Changes to Manuscript

Original Line 174:

“We assume ground-based BEVs are powered by batteries having a specific energy of under 250 Wh/kg, including technological advances (refer to VTOL Performance Enablers section for details). While lower deadweight (from reduced packaging requirements) is a benefit for

VTOLs over BEVs, battery specific energy dictates range, and has insignificant impacts on emissions from both transport modes.”

Newly added paragraph beginning Line 288:

“Fourth, lower VTOL emissions are not strongly contingent on advances in energy storage. While superior battery chemistries (and higher specific energies) favor VTOLs performance over BEVs (owing to greater weight-dependencies for the former), they affect range more than they do sustainability impacts for either transport mode, as described in the Methods section.”

Newly added paragraph beginning Line 353:

“Moreover, VTOLs would likely not be regulated by safety requirements around battery packaging as stringent as for BEVs, like those defined by the Federal Motor Vehicle Safety Standards (FMVSS).¹ Ground-based vehicles are prone to crashes and operational battery wear and tear, thereby warranting such constraints. For VTOLs, reduced overhead packaging weight enables greater realization of cell-specific to battery-specific energy compared to ground-based cars. Further, DEP enables alternative battery topographies for VTOLs, allowing for unique designs of battery warehousing. Another reason VTOLs can adopt advanced batteries earlier is that their service providers would be more likely to pay a premium compared to automotive manufacturers (from a cost-recovery and customer-base willingness to pay standpoint). The alignment of these factors indicates sufficient basis for our assumed value. ”

New paragraph also now includes a reference to the following report:

1) US DOT. Lithium-ion Battery Safety Issues for Electric and Plug-in Hybrid Vehicles. https://www.nhtsa.gov/sites/nhtsa.dot.gov/files/documents/12848-lithiumionsafetyhybrids_101217-v3-tag.pdf (2017). Accessed December 11, 2018.

Comment 13: Methods: The calculations that comprise most of the SI should be moved to Methods.

Response

We agree that having most of the SI calculations moved to the Methods section would be more helpful to the readers. Given the unlimited Methods section available to us in Nature Communications, we have transferred over a significant part of the modeling from the SI to the main paper.

Change to Manuscript

See Methods section beginning on Line 316.

Comment 1: The discussion of a high-aspect ratio blown wings enabled by DEP on lines 40 and 315 references a study of CTOL aircraft, but the conclusions of this study are not applicable to VTOL aircraft. Specifically, the cited study discusses retaining CTOL performance (e.g., a suitably low stall speed) while reducing wing area to a size better suited to efficient cruise flight; however, VTOL aircraft are, in general, not constrained to such CTOL performance considerations.

Response

We agree the Stoll et al. (2015) reference is not appropriate.

Change to Manuscript

The change to the sentence and reference at line 40 in the previous manuscript version is shown below. The limited information on DEP was replaced with a more detailed definition and the Stoll et al. (2015) reference was replaced with the Kim et al. (2018) reference. Line 37 onwards in the current version of the manuscript reads:

“One critical efficiency-enabler for VTOLs is distributed electric propulsion (DEP), which utilizes physically smaller, electrically-driven propulsors. Subsequently, propulsors can be used with greater flexibility to leverage the benefits of aero-propulsive coupling and improve performance compared to more traditional designs.¹ This enables aerodynamically optimized designs, such as articulating propellers and high aspect-ratio blown wings, which allow efficient VTOL energy performance and significant noise reduction. DEP could facilitate VTOL success in the urban aerial-taxi space, where conventional helicopters or vertical-lift aircraft have struggled.”

New reference included in the above paragraph:

- 1) Kim, Hyun D., et al. A Review of Distributed Electric Propulsion Concepts for Air Vehicle Technology, in AIAA/IEEE Electric Aircraft Technologies Symposium, 2018. Aerospace Research Central (ARC) <https://doi.org/10.2514/6.2018-4998> (2018).

Change to sentence and reference at line 315 in the previous manuscript version:

Sentence and reference have been deleted since we are now using a cruise velocity of 150 mph in accordance with the Uber (2018) whitepaper.

Comment 2: The key parameters for VTOL aircraft in Table 1 include a cruise speed of 200 mph and a cruise L/D of 17. However, while the cited Uber white paper does assume an L/D of 17, this is at 150 mph, and an L/D of 13 at 200 mph is assumed.

Response

We agree that a cruise speed of 150 mph is more appropriate for an L/D of 17 to maintain alignment with the Uber white paper.

Change to Manuscript

Please see modeling input parameters in Table 1 beginning on Line 322.

Comment 3: No rationale is given for the choice of the assumed climb L/D, which is much lower than the cruise L/D. Typically, such aircraft can reasonably be expected to climb and descend at close to their maximum L/D.

Response

The assumed climb L/D of 8 is trigonometrically related to the chosen flight path angle of 7 degrees. As stated in the Methods section, this follows our governing potential energy model, wherein climb L/D is equivalent to the ratio of horizontal slant range, and corresponding change in altitude. A diagram of the relationship is provided in Supplementary Figure 5. Thus, we calculate the climb true airspeed using the flight path angle and the rate of climb. The true airspeed needs to be reasonable (i.e. less than cruise velocity) when transitioning from vertical to horizontal flight to reduce risk. This results in a lower climb L/D compared to the cruise L/D. A lower climb L/D is also supported by the Ullman et al. (2017) study where a climb L/D of 7.5 is used for the VTOL compared to a cruise L/D of 10.

Change to Manuscript

Added Ullman et al. (2017) reference to climb L/D line item in Table 1.

Comment 4: The model and parameters given result in the climb and descent phases covering about 5 km of ground distance. It appears that this complete distance is used even when the entire flight distance is less than 5 km, resulting in disproportionately high energy usage for such shorter flights. No rationale is given for this decision. It is certainly conceivable that such an aircraft would begin its descent at a lower altitude than the stated cruise altitude, if the total flight distance is too short to reach the stated cruise altitude.

Response

We agree that distances less than 5 km were not appropriate for this analysis given the nature of the flight path. We have updated the distance domain from 1-250 km to 5-250 km, providing supporting rationale. The altitude remains fixed at 1,000 feet for all distances for controlled comparisons across scenarios.

Change to Manuscript

Original Line 113:

“Primary energy demands for each of these phases for a 1, 10, 100, and 250 km trip are shown in Figure 2.”

Changed sentence beginning Line 97:

“The base-case GHG emissions for each of these phases for trips between 5 and 250 km are shown in Figure 3. Note that a minimum of 5 km was chosen due to the 2.5 km horizontal slant range during both climb and descent phases.”

Also see updated distances in Figures 3 and 4.

Comment 5: No rationale is given for the decision to climb and descend at the flight path angle given by arctangent of the reciprocal of L/D_{climb} . Such aircraft may be more likely to climb and descend at or near their cruise airspeed, to shorten the travel times.

Response

The decision to climb and descend at the flight path angle given by arctangent of the reciprocal of the climb L/D is based on the governing potential energy model. As stated in the Methods section, a flight path angle is specified for the VTOL during climb and is trigonometrically related to the ideal climb L/D . The climb L/D is equivalent to the ratio of horizontal slant range, and corresponding change in altitude. A diagram of the relationship is provided in Supplementary Figure 5. The flight path angle is relatively small at 7 degrees, given by the arctangent of the reciprocal of climb L/D . Thus, we calculate the climb true airspeed using the flight path angle and rate of climb. Our ROC is set at 1000 fpm to avoid blade slap and follow noise abatement procedures specified by small and medium helicopter guidelines. It also would allow for both obstacle avoidance and passenger comfort.

Also note that the climb phase is a small contributor to the overall travel time for the VTOL analyzed in the baseline scenario of this study. Traveling to a cruise altitude of 1000 feet at a rate of climb of 1000 fpm means the climb phase is 1 minute in duration. Over this time, the VTOL travels 2.5 km in horizontal distance assuming the calculated ground speed of 146.4 kph. If the climb airspeed was made to be equal to the cruise airspeed as suggested, the horizontal distance traveled during climb would increase to approximately 4 km, for an increase of 1.5 km. The extra horizontal distance traveled during climb would only act to reduce the total travel time for a 100 km trip from 26.6 minutes to 25.9 minutes, or 2.6%. Therefore, the overall impact is limited.

Change to Manuscript

Please see the content contained in Methods section that was moved from the Supplementary Information to the main paper.

Reviewer 3

Comment 1: For the reader who is new to VTOL you need to differentiate them from rotorcraft also being considered as "flying cars". Pictures of VTOLs and rotorcraft may help here.

Response

We agree that a visual image of a VTOL would be helpful to readers to get a better sense of their specific design. We have added a referenced public domain image of a prototype NASA VTOL in takeoff, hover, and landing mode. Most demonstration VTOLs have image copyrights we do not have access to, which limited our ability to add even more pictures.

Change to Manuscript

Please refer to Figure 1 in revised manuscript.

Comment 2: Line 61: Need to spell out ICEVs and BEVs on first use for the reader new to the field.

Response

We agree that spelling out those acronyms would be helpful to new readers. Please see the addition to the manuscript below.

Change to Manuscript

Original paragraph beginning line 61:

“Our physics-based models indicate that for 100 km point-to-point trips VTOL GHG emissions are roughly 30% lower and 40% higher than ICEVs and BEVs, respectively.”

New paragraph with definition beginning line 81:

“Our physics-based models indicate that for the base-case with 100 km point-to-point trips, VTOL GHG emissions are 36% lower than internal combustion engine vehicles (ICEVs), but 27% higher than battery electric vehicles (BEVs).”

Comment 3: Figure 1 in the manuscript is messed up. S1 is correct.

Response

We have updated Figure 1 (now Figure 2) accordingly.

Change to Manuscript

Please see the updated version of Figure 2.

Comment 4: What is "S1" and figures "S5", "S6" etc? I figured out that they referred to the supplemental material. I guess I am not familiar with Nature Communications' handling of papers and support material. Should the reader be told this or will the links be active in the final paper?

Response

We agree that it may not be intuitive to the reader what the ‘S’ represents in any of the Figures referenced to the SI. Hence, in all instances where we refer to said figures, we preface with the word “Supplementary”.

Change to Manuscript

References to SI now explicitly mention “Supplementary”. For instance:

Updated sentence beginning Line 104:

“Primary energy results follow the same trends as GHG emissions and are reported in Supplementary Figures 1 and 2.”

Comment 5: Figures in your sensitivity analysis section would be helpful for the reader.

Response

We agree that a more visual depiction of the sensitivity analyses would be helpful to the reader. The parameters being varied (with corresponding bounds) have been explicitly mentioned in Table 1, with subsequent sensitivity results shown in Figure 5.

Change to Manuscript

Please see updated version of Table 1 and Figures 5a and 5b.

Comment 6: Note that I have not checked the equations. I assume the authors have compared their result to those of German and Ullman for verification. They need to state this or other support for why I should believe their results.

Response

We agree that comparing our results to the German (Uber) and Ullman et al. studies would provide helpful verification. Please see the addition to the manuscript below.

Change to Manuscript

New paragraph with verification beginning line 293:

“Related work by Uber and Ullman et al. support the key takeaways from our study. Uber estimates a VTOL energy intensity of about 0.48 kWh/km at 241 kph for an 80-km trip. No detailed breakdown of the VTOL energy modeling is provided. Under these conditions and assuming four occupants, the VTOL operational energy intensity yielded by our model is 0.43 kWh/km (about 10% lower). Further, the Ullman et al. model of VTOL range and energy consumption was reproduced according to the physics-based relationships stated in the study. Using Ullman et. al.’s model and input assumptions, the VTOL operational energy intensity is 0.57 kWh/km for a 1360 kg VTOL at 241 kph for a 100-km trip. Using our baseline input assumptions instead, Ullman’s model produces an energy intensity of 0.36 kWh/km (about 36% lower), which is the same as the output from our model for a fully-loaded VTOL. Despite these studies reporting higher energy intensities, the comparison to ground-based vehicles still remains promising. Using the 2020 U.S. average grid mix, Uber’s 0.48 kWh/km and Ullman’s 0.57 kWh/km translate to GHG emissions of 0.09 kg-CO₂e/PKT and 0.10 kg-CO₂e/PKT, respectively (for 3 passengers on a 100 km trip). These compare favorably to the 0.07 kg-CO₂e/PKT and 0.13 kg-CO₂e/PKT results for the BEV and ICEV, respectively.”

Comment 7: Why no uncertainty bars on the BEVs in Fig S6?

Response

Figure S6 has been replaced with Figure 4 in the manuscript for GHG emissions and Supplementary Figure 3 for primary energy. The format of the Figure has been changed, and uncertainties are now explored in the sensitivity analysis (Supplementary Figure 4).

Change to Manuscript

N/A

Comment 8: Be nice to end with some conclusions (restated from the abstract) and next steps. The paper as it is now, just stops.

Response

The format requirements for Nature Communications prohibits a conclusion section. Instead, we put forth the key takeaways in the Discussion section, then end the paper with the Methods section as required.

Change to Manuscript

Please see Discussion section beginning on Line 252.

Reviewers' comments:

Reviewer #1 (Remarks to the Author):

The authors have thoroughly responded to my comments on the previous draft. I recommend this article for publication.

Reviewer #2 (Remarks to the Author):

I found the revised manuscript to be generally well-executed and technically accurate, with a few exceptions:

1. The system efficiency factor η accounts for the combination of electrical losses in the propulsion system and aerodynamic losses in the propeller. However, the hover figure of merit is employed in addition to this factor to account for aerodynamic losses in hover, resulting in a double-bookkeeping of the aerodynamic losses in hover. Perhaps the aerodynamic and electrical inefficiencies could be bookkept separately for all phases of flight to avoid this issue. Note that reference 22 (Brown and Harris 2018) describes a powertrain efficiency of 90% and a propulsive efficiency of 85% for a total efficiency of $0.9 \times 0.85 = 76.5\%$, not the value of 85% given in this manuscript; also, the value given from reference 21 (Stolaroff 2018) describes a quadcopter and is not applicable to the VTOL aircraft considered in this manuscript.

2. The maximum takeoff mass is given as 1187.5 kg, but on line 442 this same figure is given as the takeoff mass with a single occupant (pilot), which must be less than the maximum takeoff mass to allow payload mass for air taxi missions. With a full load of 4 passengers, the takeoff weight is only 1450 kg ($1187.5 \text{ kg} + 3 \times 87.5 \text{ kg}$), which is low compared to the values of 1800 kg in reference 2 (Datta 2018, table 2) and 4000 lb (1814 kg) in references 7 (Uber 2016) and 16 (Stoll 2015) for similar 4-passenger electric VTOL aircraft.

3. While the takeoff weight with four passengers is 1450 kg, the disk area of the 1814 kg aircraft in reference 16 (Stoll 2015) is used, which is somewhat incongruous. A more reasonable approach would be to employ the disk loading given in reference 16, rather than the disk area. I did not see how the rotor blade area affects the results.

4. I did not find the rationale for the climb conditions to be sensible. The L/D in climb should be a function of only the aircraft aerodynamic performance, governed for a particular aircraft by its airspeed and lift, and is independent of the flight path angle. For an aircraft and mission like those considered herein, it is reasonable to assume the airspeed in the climb portion will be equal to the cruise airspeed (to minimize the block time of the mission), in which case the L/D in climb will be approximately equal to the L/D in cruise (since the lift in these segments differs only by a factor of the cosine of the climb angle, which is approximately 1 for reasonable climb angles).

I have two concerns with the climb L/D of 8 given by the cited reference: the first is that, in the aircraft described in the associated reference (Ullman), the cruise L/D is only 10, so the given climb L/D of 7.5 means that the climb power is about 133% the cruise power. However, in the case described in this manuscript, the cruise L/D is 17, but a climb L/D of 8 is assumed, meaning that the climb power is about 213% the cruise power. If the climb performance were to be based off of this reference, it would be more appropriate to use the same 133% factor. However, the second concern is that the rationale given in the cited paper for the climb L/D is only, "There are no good sources of L/D values for climb so the value of 10 seems generous." It appears the authors of the cited paper didn't know what L/D value in climb is appropriate, so apparently made something up with little basis. As stated above, it would be reasonable to use the same airspeed and L/D in climb as in cruise.

5. While the given climb rate of 1000 ft/min is a reasonable value for such an aircraft and mission,

the rationale given for this climb rate is based on its applicability to helicopters (although the cited reference mostly concerns itself with blade slap and noise abatement concerns in descent, and, in a brief perusal, I did not see recommendations for specific climb rates, beyond simply suggesting higher climb rates are more desirable), which does not necessarily extend to such VTOL aircraft (which will likely perform climb and descent in airplane mode, where blade slap, etc. are not concerns).

Reviewer #3 (Remarks to the Author):

I have read in detail the responses and they address the issues raised. I have not reread the manuscript in detail, but scanned it and it appears much stronger and suitable for publication.

We would like to thank the editor and the reviewers for their valuable feedback and comments. Based on reviewer comments, we have made further changes to the manuscript that enhance its overall quality. Please find our responses to the comments below.

Reviewer 2

Comment 1: The system efficiency factor η accounts for the combination of electrical losses in the propulsion system and aerodynamic losses in the propeller. However, the hover figure of merit is employed in addition to this factor to account for aerodynamic losses in hover, resulting in a double-bookkeeping of the aerodynamic losses in hover. Perhaps the aerodynamic and electrical inefficiencies could be booked separately for all phases of flight to avoid this issue. Note that reference 22 (Brown and Harris 2018) describes a powertrain efficiency of 90% and a propulsive efficiency of 85% for a total efficiency of $0.9 \times 0.85 = 76.5\%$, not the value of 85% given in this manuscript; also, the value given from reference 21 (Stolaroff 2018) describes a quadcopter and is not applicable to the VTOL aircraft considered in this manuscript.

Response

We agree that the aerodynamic losses were previously double-booked during hover. We have corrected this by consistently defining system efficiency as being composed of powertrain efficiency and propulsive efficiency, and defining separate climb/cruise (η_c) and hover (η_h) system efficiency values. The system efficiency for climb and cruise is 0.9 (powertrain) * 0.85 (propulsive) = 0.765 according to Brown and Harris (2018). The new bounds for the parameter sensitivity analysis are set at 0.7 and 0.8 ($\pm 5\%$ convention). The system efficiency for hover is 0.9 (powertrain) * 0.7 (propulsive) = 0.63 according to Ullman (2017).

Change to Manuscript

Added the following sentence beginning at Line 380:

“ η is composed of powertrain efficiency (0.9) and propulsive efficiency (0.85 for climb and cruise, and 0.7 for hover).”

Added the following sentence beginning at Line 442:

“In addition to disk loading (δ), we use a hover system efficiency (η_h) of 0.63 , which incorporates a powertrain efficiency of 0.9 and propulsive efficiency of 0.7 (instead of 0.85) to account for lifting inefficiencies.”

Modified the following sentence beginning at Line 478:

“Combining these two power elements yields Equation (11), which also incorporates climb system efficiency (η_c).”

Modified the following sentence beginning at Line 511:

“This yields Equation (14), which also considers cruise system efficiency (η_c).”

Updated Equation 8 to remove the double-booked hover figure of merit:

$$P_{\text{hover}} = \frac{Wg}{\eta_h} \sqrt{\frac{\delta}{2\rho}}$$

Change to Table 1 beginning Line 326:

Separated system efficiency into climb/cruise and hover values.

Comment 2: The maximum takeoff mass is given as 1187.5 kg, but on line 442 this same figure is given as the takeoff mass with a single occupant (pilot), which must be less than the maximum takeoff mass to allow payload mass for air taxi missions. With a full load of 4 passengers, the takeoff weight is only 1450 kg (1187.5 kg + 3*87.5 kg), which is low compared to the values of 1800 kg in reference 2 (Datta 2018, table 2) and 4000 lb (1814 kg) in references 7 (Uber 2016) and 16 (Stoll 2015) for similar 4-passenger electric VTOL aircraft.

Response

We agree that 1187.5 kg should be consistently reported as the takeoff mass with a single occupant (pilot), with 1450 kg being the maximum takeoff mass for a fully-loaded VTOL (four occupants).

Our ratio of VTOL structural mass (M_s) to maximum takeoff mass (i.e., 0.51) was an intermediate value selected from the range of airframe fraction values defined in Ullman (2017), which varies from 0.44 to 0.55. Further, very similar to our values, Datta (2018) lists the Napoleon Aero VTOL with a maximum takeoff mass of 1500 kg, and a maximum payload of 400 kg. The same report also mentions the Bartini Flying Car (400 kg maximum payload) which has a *lower* maximum takeoff mass (1100 kg) than our VTOL. Next, part of the difference from the Uber values is attributed to them using a significantly higher payload mass, 1100 lb, (500 kg) reported in Uber (2018). The Stoll (2015) specifications are based on battery-packs available at the time of publication, which were likely of the order of 250 Wh/kg or lower – implying that their reported maximum takeoff mass would naturally be higher due to the higher battery mass. Finally, Brown and Harris (2018) optimize for takeoff mass and conclude that there is a large variability in results (2800 – 4000 lb for similar trips) depending on the mission, technical assumptions, and vehicle requirements, especially battery specific energy, range, automation level, and reserve requirements. Hence, our

takeoff mass of 1187.5 kg for a single occupant and 1450 kg for four occupants seems reasonable when compared to the existing literature.

To that end, we have now reworded “maximum takeoff mass” to “takeoff mass” across the manuscript for avoiding any confusion, and have also updated the reference for structural mass in Table 1 to *Calculated & Ullman et al., 2017*.

Change to Manuscript

Please see Table 1 (beginning Line 326) for updated nomenclature of takeoff mass and reference for structural mass.

Comment 3: While the takeoff weight with four passengers is 1450 kg, the disk area of the 1814 kg aircraft in reference 16 (Stoll 2015) is used, which is somewhat incongruous. A more reasonable approach would be to employ the disk loading given in reference 16, rather than the disk area. I did not see how the rotor blade area affects the results.

Response

We agree that the takeoff weight and disk area were incongruous. We have adopted the suggestion to employ the disk loading of 450 N/m² from Stoll (2015).

Change to Manuscript

Original paragraph beginning on Line 435:

“Hover power is primarily dependent on rotor disk loading, defined as the VTOL total weight divided by the lifting surface area. **Equation (8)** demonstrates the tradeoffs VTOL designers will have to make, wherein larger rotors makes the VTOL more efficient in hover but can compromise cruise performance and be significantly noisier with higher rotor tip speeds. The disk loading parameter is chosen based on specifications for a tilt rotor VTOL with six (N) 1.45-meter diameter rotors.¹⁶ This equates to an individual rotor blade area (A_b) of 6.6 m² and a total rotor lifting area of about 40 m², resulting in a δ value of 294.2 N/m² for the VTOL. For a single occupant (pilot), the VTOL takeoff mass W is 1187.5 kg. Hence:

$$\delta = \frac{Wg}{A} = \frac{Wg}{NA_b} = \frac{1187.5 \text{ kg} \cdot 9.81 \frac{\text{m}}{\text{s}^2}}{6 \cdot 6.6 \text{ m}^2} = 293.9 \text{ N/m}^2 \quad (9)''$$

New paragraph beginning on Line 439:

“Hover power is primarily dependent on rotor disk loading, defined as the VTOL total weight divided by the lifting surface area. The disk loading parameter is chosen based on data provided in Stoll (2015), resulting in a δ value of 450 N/m² for the VTOL.²¹”

Comment 4: I did not find the rationale for the climb conditions to be sensible. The L/D in climb should be a function of only the aircraft aerodynamic performance, governed for a particular aircraft by its airspeed and lift, and is independent of the flight path angle. For an aircraft and mission like those considered herein, it is reasonable to assume the airspeed in the climb portion will be equal to the cruise airspeed (to minimize the block time of the mission), in which case the

L/D in climb will be approximately equal to the L/D in cruise (since the lift in these segments differs only by a factor of the cosine of the climb angle, which is approximately 1 for reasonable climb angles). I have two concerns with the climb L/D of 8 given by the cited reference: the first is that, in the aircraft described in the associated reference (Ullman), the cruise L/D is only 10, so the given climb L/D of 7.5 means that the climb power is about 133% the cruise power. However, in the case described in this manuscript, the cruise L/D is 17, but a climb L/D of 8 is assumed, meaning that the climb power is about 213% the cruise power. If the climb performance were to be based off of this reference, it would be more appropriate to use the same 133% factor. However, the second concern is that the rationale given in the cited paper for the climb L/D is only, “There are no good sources of L/D values for climb so the value of 10 seems generous.” It appears the authors of the cited paper didn’t know what L/D value in climb is appropriate, so apparently made something up with little basis. As stated above, it would be reasonable to use the same airspeed and L/D in climb as in cruise.

Response

We agree that the rationale for our selection of the climb L/D could be more robust. We have addressed the comment in a point-by-point format below and added a consolidated form of the rationale in the manuscript, complete with a new reference and a parameter-sensitivity for climb L/D.

I did not find the rationale for the climb conditions to be sensible.

The fundamentals of electric aircraft performance support our rationale for the climb conditions. Potential energy and the Breguet range equation for electric aircraft give us the assumption that L/D in climb is related to the climb rate of the aircraft. Range of an electric aircraft is given as follows:

$$R = E \frac{L}{D} \eta \frac{1}{Wg} \quad (3)$$

Potential energy describes the energy requirement for an aircraft to get to a given altitude:

$$E = \frac{Wgh}{\eta} \quad (1)$$

Combining these fundamentals, we get the following equation which relates the aerodynamic efficiency of the aircraft (L/D) to the trigonometry of flight:

$$\frac{L}{D} = \frac{R}{h} \quad (2)$$

This relation means quite simply, an aircraft climbing to a specified altitude (h) will have a resultant ground roll, or horizontal slant range (R) based on the aerodynamic performance of the aircraft (L/D). This assumption is a standard practice for aviation. All aircraft have an associated Pilot Operations Handbook (POH) which describes in detail the associated ground roll of an aircraft climbing to a specified altitude at a specified speed which is directly related to the aircraft aerodynamic performance.

The same relationship can be used for an aircraft in descent. An aircraft flying at a given altitude has an associated amount of potential energy. Altitude is directly related to the distance an aircraft can travel based on the aerodynamic performance. Again, this is a standard practice in aviation. A pilot executing an unpowered glide understands that their altitude corresponds to the ground distance they can travel based on the aerodynamic performance of the airplane. In an emergency descent, a pilot will choose a landing site with their associated altitude in mind.

The L/D in climb should be a function of only the aircraft aerodynamic performance, governed for a particular aircraft by its airspeed and lift, and is independent of the flight path angle.

We agree that the L/D ratio is a direct function of the aircraft speed. However, as demonstrated above, we see a fundamental relationship between the L/D ratio and the trigonometry of flight.

For an aircraft and mission like those considered herein, it is reasonable to assume the airspeed in the climb portion will be equal to the cruise airspeed (to minimize the block time of the mission), in which case the L/D in climb will be approximately equal to the L/D in cruise (since the lift in these segments differs only by a factor of the cosine of the climb angle, which is approximately 1 for reasonable climb angles).

We disagree that climb for our VTOL should be modeled at cruise airspeed. We believe climb should be modeled at a slower true airspeed than cruise, for two reasons. First, a VTOL will realistically have to transition non-instantaneously from hover to cruise while climbing. In hover, the configuration of the aircraft will likely be much different than in cruise. The aircraft will initially have zero ground speed starting from hover. This means the VTOL will have to transition to an airplane configuration and accelerate all the way to cruise true airspeed by the time it reaches cruise altitude. Because the VTOL climbs to a shallow cruise altitude of 1000 ft, this transition process cannot be disregarded as an exceptionally small part of climb. For this reason, we assume that the average climb true airspeed will be lower than the cruise true airspeed.

Second, we expect that a VTOL will ideally climb at the best rate of climb airspeed (v_y). Best rate of climb is the true airspeed that allows an aircraft to climb to an altitude in the shortest amount of time possible. This airspeed will be lower than cruise true airspeed. Our model assumes a slower climb velocity, consistent with the best rate of climb, because this still minimizes the block time of the mission spent in climb. This is important for several reasons. A climbing aircraft is more uncomfortable to passengers than an aircraft in cruise. Time spent in climb should be minimized to account for passenger comfort. Climbing over obstacles is another important consideration. Obstacles include buildings in a congested urban setting but also the FAA mandated separation distance around them. While

theoretically a VTOL could climb vertically all the way to cruise altitude, this would be neither efficient nor fast. Climbing near v_y will ensure both a more efficient and faster climb to altitude.

Time near the ground should also be minimized as much as possible. Noise abatement for practical VTOL operation will be of the utmost importance, despite not being a point of research for our model. The sound intensity is inversely proportional to the square of the distance. A VTOL at higher altitudes will be significantly quieter. Therefore, the rate at which a VTOL climbs away from daily life on the ground should be maximized. Thus, our model assumes a lower climb airspeed than cruise airspeed.

Our rationale was further confirmed by two subject matter experts we conversed with. William Fredericks, CEO of Advanced Aircraft Company and ex-NASA Langley VTOL researcher communicated that “It depends on how the user will operate the aircraft. If the aircraft is flying much faster in cruise, then no, that would be a bad assumption to use the same L/D in both [climb and cruise] conditions”. Dr. Carlos Cesnik, AIAA Deputy Director for Structures and University of Michigan Professor of Aerospace Engineering said “The L/D will depend on configuration choices and considering the significant change in configuration for a VTOL aircraft between climbing and cruise, they should be different. I would be very surprised if you can get even half of your currently assumed cruise L/D during climbing”. Both of them have subsequently been added to the Acknowledgments section of our manuscript.

I have two concerns with the climb L/D of 8 given by the cited reference...

We agree that the prior Ullman reference does not provide enough basis for a L/D of 8 in climb. Instead we now relate the L/D of our VTOL model to the performance of the Cirrus SR-22, the world’s most widely-produced general aviation aircraft over the past two decades. Below is Figure 5 from “Conceptual Design of High-Lift Propeller Systems for Small Electric Aircraft” by Dr. Michael Patterson of Georgia Institute of Technology Aerospace Engineering, guided and approved of Dr. Brian German, VTOL SME, whose work we have previously cited. We propose this new reference for supporting a climb L/D value of 8.

This figure shows the relationship between L/D and true airspeed for different configurations (maximum lift coefficients) of an aircraft with aerodynamics similar to the SR-22. In agreement with our model, we will look specifically at the maximum lift coefficient value of 3 because it coincides with the expected aerodynamic performance of our VTOL. That is, we specify that our VTOL will cruise at the maximum L/D in order to maximize the efficiency of the mission. For $C_{L,max}$ of 3, the maximum L/D occurs at our intended cruise speed of 130 knots (150 mph). Note that alternatively we could specify a higher cruise speed at the expense of aerodynamic performance. This is often the case in general aviation in which pilots will cruise at a higher speed on the lower end of the L/D curve to minimize block time of the mission. However, because of the range limitations of electric aircraft, it is ideal to assume aerodynamic performance will be optimized over mission block-time.

Further analyzing the curve for a $C_{L,max}$ of 3, we note that the max L/D is about 15, which is lower than our specified value of 17. Despite a VTOL design being fundamentally different than a general aviation aircraft to account for vertical takeoff and landing, we expect their aerodynamics to be very similar. Electrification of VTOLs will have inherent performance benefits over traditional single engine aircraft. Additionally, aircraft such as the SR-22 have oversized wings to account for minimum stall speed requirements which limits their aerodynamic performance at higher speeds. We can expect that an electric VTOL will have a higher L/D max than a general aviation airplane. With this in mind, we can use the above performance data to establish a ratio we can transpose and apply to our model.

We assume the VTOL begins with a ground speed of 0 in hover and smoothly accelerates to its cruise speed of 130 knots over 1 minute. This yields an average velocity of 65 knots. For $C_{L,max} = 3$, a 65-knot average velocity corresponds to a L/D of approximately 7. This yields a ratio of 214% between the maximum L/D of 15 and the average cruise L/D of 7. Note that our approximation can be justified owing to the relatively linear nature of this segment of the curve. Applying this ratio to our cruise L/D of 17 results in a climb L/D of 8. This provides a more substantive and robust justification for our selection of the baseline climb L/D being 8. However, to consider the scenario where a VTOL might have identical climb and cruise

velocities, wherein changing climb L/D from 8 to 17 (equal to cruise), we now conduct a parameter sensitivity analysis for climb L/D. We find that changing this input from 8 to 17 alters overall results by <5%. Figure 5a in the manuscript has been updated accordingly.

Change to Manuscript

Added the following paragraph beginning on Line 464:

“ L/D_{climb} is expected to be lower than L/D_{cruise} due to the lower average true airspeed during climb. The VTOL will begin climb with a ground speed of 0 mph and smoothly accelerate to the 150 mph cruise true airspeed over the 1 minute climb phase, resulting in an average climb true airspeed of 75 mph. The L/D_{cruise} to L/D_{climb} ratio corresponding to these cruise and climb airspeeds is 2.14 according to the modeling of an aircraft with similar aerodynamics to a Cirrus SR-22.¹ Applying this factor to the L/D_{cruise} of 17 produces a L/D_{climb} of 8. The sensitivity analysis indicates the overall results are not strongly dependent on the selection of the L/D_{climb} , varying by less than 5% when the L/D_{climb} is set equal to L/D_{cruise} with a value of 17.”

Additional reference added:

1. Patterson, M. Conceptual Design of High-Lift Propeller Systems for Small Electric Aircraft. <https://smartech.gatech.edu/bitstream/handle/1853/55569/PATTERSON-DISSERTATION-2016.pdf> (2016). Accessed January 19, 2019.

Change to Table 1 beginning Line 326:

Removed “Ullman et al., 2017” reference for the climb L/D and replaced with “Patterson, 2016”.

Comment 5: While the given climb rate of 1000 ft/min is a reasonable value for such an aircraft and mission, the rationale given for this climb rate is based on its applicability to helicopters (although the cited reference mostly concerns itself with blade slap and noise abatement concerns in descent, and, in a brief perusal, I did not see recommendations for specific climb rates, beyond simply suggesting higher climb rates are more desirable), which does not necessarily extend to such VTOL aircraft (which will likely perform climb and descent in airplane mode, where blade slap, etc. are not concerns).

Response

We agree that a better alternative reference should be used to back our chosen climb rate of 1000 fpm, and a rationale applicable to noise abatement/helicopters should be avoided for our mission.

A 2016 Joby report by Stoll and Mikic, pertinent to DEP commuter aircraft, is now cited instead. Note that the same climb rate is also specified by the Opener BlackFly eVTOL and the Dzyne Technologies Inc fixed wing VTOL. Further, the rationale alluding to steep climb rates and blade slap have been eliminated. Please see the changes to the manuscript below.

Change to Manuscript

Original Line 458:

“Our ROC is set at 1000 fpm to avoid blade slap and follow noise abatement procedures specified by small and medium helicopter guidelines.¹ This climb rate would allow for both obstacle avoidance and passenger comfort.”

Original Reference:

1. Cox , C., et al. Fly neighborly guide. Helicopter Association International. Technical Report. 2009.

Changed sentence beginning Line 456:

“The ROC for our VTOL mission is set at 1000 fpm.¹ This climb rate would allow for both obstacle avoidance and passenger comfort.”

Updated Reference:

1. Stoll, A. M., & Veble Mikic, G. (2016). Design studies of thin-haul commuter aircraft with distributed electric propulsion. In 16th AIAA Aviation Technology, Integration, and Operations Conference (p. 3765).

Deleted Original Line 584:

“The ROC is lower than for commercial aircraft, and yet enables avoidance of noisy blade slap.”

Reviewers' comments:

Reviewer #2 (Remarks to the Author):

Although fairly inconsequential to these results, the calculation of climb performance given here involves a fundamental misunderstanding of the relevant physics that has no place in a publication such as this. The range equation $R = E \cdot (L/D) \cdot \eta / W$ (where "W" here more sensibly represents the vehicle weight, rather than its mass) describes the energy required to counteract aerodynamic drag over a distance R; the equation $E = Wh$ describes the potential energy associated with an altitude difference h. The equation $L/D = R/h$, then, implies only that the ratio of the energy required to counteract drag over a flight distance to the potential energy of an altitude difference is equal to the aircraft's L/D. The energy usage during a climb is equal to the sum of the energy required to counteract drag over the flight distance and energy required to increase the gravitational potential energy: $E = (R \cdot W / (L/D) + W \cdot h) / \eta$. In this context, the value of L/D can be assumed to be a function of only the lift coefficient (CL) the aircraft is flying at, which is approximately given by $CL = W / (0.5 \cdot \rho \cdot V^2 \cdot S)$. Therefore, for a given aircraft weight, the L/D is approximately a function of only the equivalent airspeed (and independent of climb/descent conditions).

Additionally, the authors assume that acceleration and deceleration to and from hover occur during the climb and descent phases; however, the assumption that the energy usage during these segments can be approximated by applying the L/D calculated at the average speed during the respective segment is not a valid assumption. This is primarily because the L/D curve of cruise flight does not apply to lower-speed flight, when the VTOL aircraft is partially wing-borne and partially thrust-borne, and in addition, may experience additional drag resulting from extended flaps and/or landing gear. While I appreciate the ambitious intent to model the effects of climb/acceleration and descent/deceleration, these segments do not lend themselves to simplified modeling (due to the complex and varied performance of different VTOL aircraft configurations during the transition regime), and in my view, any model describing performance in this regime of sufficient accuracy to be worth using would also be of a complexity beyond the scope of this effort. My recommendation is to ignore the climb/acceleration and descent/deceleration effects entirely, and model the flight as only takeoff hover, cruise, and landing hover segments. The energy required in excess of cruise performance to climb and accelerate is approximately balanced out by the lower energy required during the descent and deceleration segment, such that assuming cruise performance for the whole duration is a good approximation. While the decrease in L/D at lower speeds (during acceleration/deceleration to/from cruise speed) would not be modeled, the current assumption of steady hover for 30 seconds at each end of the mission overpredicts energy usage for these segments, which should approximately compensate. (The reason this is an overprediction is that a realistic takeoff or landing trajectory would not loiter in hover, but spend more time with forward speed, where power requirements are lower due to translational lift on the rotors and/or wing lift.) This simplified model will be no less accurate than the current modeling assumptions, but its underlying assumptions and simplifications are more reasonable.

Alternatively, energy used in the complex acceleration and deceleration segments could reasonably be assumed to be approximately covered by the present hover modeling (i.e., 30 seconds of steady hover at each end of the flight), and the climb and descent could be modeled as taking place at the cruise speed of 150 mph and its associated L/D of 17. However, the total energy usage calculated in such a way will not differ significantly from the total energy usage calculated with the previous suggestion of assuming only cruise power over the entire flight distance, so the trivial increase in fidelity may not be worth the additional modeling complexity.

If the authors do prefer to extrapolate the cruise L/D curve to lower speeds, the Cirrus SR22 reference is not necessary, given the L/D figures already provided by the Uber reference. (The SR22 reference is also less relevant, since the Uber values are more representative of such a VTOL air taxi aircraft than those derived from the SR22). The widely-accepted two-parameter quadratic drag model can be developed for this aircraft from the two L/D values given in the Uber reference—17 at 150 mph and 13 at 200 mph—resulting in the curve $L/D = 1 / [(V / 790 \text{ mph})^2 + (22.6 \text{ mph} / V)^2]$, where V is the equivalent airspeed in mph.

We would like to thank the editor and the reviewer for their valuable feedback. Based on the reviewer's comment, we have made further changes to the manuscript that enhance its overall quality. Please find our responses to the comment below, which we have addressed in a point-by-point format. We have added a consolidated form of the rationale in the manuscript.

Reviewer 2

Comment Part 1: Although fairly inconsequential to these results, the calculation of climb performance given here involves a fundamental misunderstanding of the relevant physics that has no place in a publication such as this. The range equation $R = E \cdot (L/D) \cdot \eta / W$ (where "W" here more sensibly represents the vehicle weight, rather than its mass) describes the energy required to counteract aerodynamic drag over a distance R; the equation $E = Wh$ describes the potential energy associated with an altitude difference h. The equation $L/D = R/h$, then, implies only that the ratio of the energy required to counteract drag over a flight distance to the potential energy of an altitude difference is equal to the aircraft's L/D.

The energy usage during a climb is equal to the sum of the energy required to counteract drag over the flight distance and energy required to increase the gravitational potential energy: $E = (R \cdot W / (L/D) + W \cdot h) / \eta$. In this context, the value of L/D can be assumed to be a function of only the lift coefficient (CL) the aircraft is flying at, which is approximately given by $CL = W / (0.5 \cdot \rho \cdot V^2 \cdot S)$. Therefore, for a given aircraft weight, the L/D is approximately a function of only the equivalent airspeed (and independent of climb/descent conditions).

Response

We agree that the present notation for takeoff mass (W) can be confusing to the reader, and have updated the symbol for takeoff mass to "m", with VTOL weight "W" now defined as the product of "m" and acceleration due to gravity "g". All relevant equations have been subsequently modified.

Further, the climb energetics equation stated here by the reviewer, $E = (R \cdot W / (L/D) + W \cdot h) / \eta$, is identical to Equation (12) stated in the previous manuscript when normalized for time (energy v/s power; height v/s velocity). As stated, this equation is comprised of energy requirements for both trip length and potential energy. Also, we continue to agree that for a given aircraft weight, the L/D is approximately a function of only the equivalent airspeed.

Change to Manuscript

Notation for takeoff mass has been changed across the manuscript from “W” to “m”; VTOL weight “W” has been defined as the product of mass “m” and acceleration due to gravity “g”. All references to these variables have been subsequently updated.

Comment Part 2: Additionally, the authors assume that acceleration and deceleration to and from hover occur during the climb and descent phases; however, the assumption that the energy usage during these segments can be approximated by applying the L/D calculated at the average speed during the respective segment is not a valid assumption. This is primarily because the L/D curve of cruise flight does not apply to lower-speed flight, when the VTOL aircraft is partially wing-borne and partially thrust-borne, and in addition, may experience additional drag resulting from extended flaps and/or landing gear. While I appreciate the ambitious intent to model the effects of climb/acceleration and descent/deceleration, these segments do not lend themselves to simplified modeling (due to the complex and varied performance of different VTOL aircraft configurations during the transition regime), and in my view, any model describing performance in this regime of sufficient accuracy to be worth using would also be of a complexity beyond the scope of this effort.

Response

We agree that modeling climb energetics using an average speed, and transposing the cruise L/D curve to lower-speed climb might not be appropriate assumptions. We have now stated in the manuscript that modeling the climb phase of the flight without having an accurate velocity profile and specific aircraft configuration prevents us from using our high-level, physics-based approach. Further, we have deleted the rationale for using an average speed of 75 mph in climb, alongside omitting the reference to the climb L/D parameter.

Change to Manuscript

Original Line 389:

“The VTOL energy model considers each flight mode separately: hover, climb, cruise, descent, and reserves.”

Changed sentence beginning Line 390 (‘clean’ manuscript):

“The VTOL energy model combines climb and descent with cruise due to the uncertainty in the speed profile and transition to/from winged flight during these phases.”

Deleted text beginning Line 460:

“A flight path angle (γ) is specified for the VTOL during climb. and is trigonometrically related to the ideal climb ratio (L/D_{climb}). This follows our governing potential energy model, wherein L/D_{climb} is equivalent to the ratio of horizontal slant range, and corresponding change in altitude. A diagram of the relationship is provided in Supplementary Figure 5. L/D_{climb} is expected to be lower than L/D_{cruise} due to the lower average true airspeed during climb. The VTOL will begin climb with a ground speed of 0 mph and smoothly accelerate to the 150 mph cruise true airspeed over the 1 minute climb phase,

resulting in an average climb true airspeed of 75 mph. The L/D_{cruise} to L/D_{climb} ratio corresponding to these cruise and climb airspeeds is 2.14 according to the modeling of an aircraft with similar aerodynamics to a Cirrus SR-22.¹⁸ Applying this factor to the L/D_{cruise} of 17 produces a L/D climb of 8. The sensitivity analysis indicates the overall results are not strongly dependent on the selection of the L/D_{climb}, varying by less than 5% when the L/D_{climb} is set equal to L/D_{cruise} with a value of 17.

The flight path angle (γ) is relatively small at 7 degrees, given by the arctangent of the reciprocal of L/D_{climb}.”

Deleted Original Reference 18:

18) Patterson, M. Conceptual Design of High-Lift Propeller Systems for Small Electric Aircraft. <https://smartech.gatech.edu/bitstream/handle/1853/55569/PATTERSON-DISSERTATION-2016.pdf> (2016). Accessed January 19, 2019.

Added paragraph beginning Line 450 ('clean' manuscript):

“The climb and descent phases are modeled in the same way as cruise for three main reasons. First, the energy required in excess of cruise performance to climb and accelerate is approximately balanced out by the lower energy required during the descent and deceleration segment, such that assuming cruise performance for the whole duration is a good approximation. Second, limited data are available indicating how the VTOL true air speed and corresponding L/D would change throughout climb and descent. Finally, due to the cruise altitude of 1000 ft and the assumed rate of climb (ROC) and rate of descent (ROD) of 1000 fpm, the climb and descent phases have a duration of only 2 minutes, which is only a small portion of the 25-minute flight in the base case.”

Modified paragraph beginning Line 459 ('clean' manuscript):

“However, if/when an accurate velocity profile and VTOL configuration is made available, a higher fidelity modeling approach may be used. In this case, climb would be modeled separately from cruise, and power requirements would be split up into two distinct parts. First, the potential energy used to lift the VTOL to a given altitude is converted to power ($P_{\text{climb,PE}}$) by dividing by the time for climb (t_{climb})... [Next, we consider the power necessary to overcome aerodynamic forces during climb...]”

Added Line 475 ('clean' manuscript):

“This yields the final power Equation (11) for climb (P_{climb}), which would have to be integrated over the velocity and L/D profile during the climb phase.”

Comment Part 3: My recommendation is to ignore the climb/acceleration and descent/deceleration effects entirely, and model the flight as only takeoff hover, cruise, and landing hover segments. The energy required in excess of cruise performance to climb and accelerate is approximately balanced out by the lower energy required during the descent and deceleration segment, such that assuming cruise performance for the whole duration is a good approximation. While the decrease in L/D at lower speeds (during

acceleration/deceleration to/from cruise speed) would not be modeled, the current assumption of steady hover for 30 seconds at each end of the mission overpredicts energy usage for these segments, which should approximately compensate. (The reason this is an overprediction is that a realistic takeoff or landing trajectory would not loiter in hover, but spend more time with forward speed, where power requirements are lower due to translational lift on the rotors and/or wing lift.) This simplified model will be no less accurate than the current modeling assumptions, but its underlying assumptions and simplifications are more reasonable.

Response

We agree that it would be reasonable here to combine the energy expended in climb/descent with the cruise phase, instead of parsing out their individual contributions. The rationale of our conservative hover phase overestimating energy, and thus compensating for us ignoring the lower performance (and L/D) at reduced speeds has been added to the manuscript. The assumption of excess energy expended to climb and accelerate balancing out the lower energy needed for deceleration and descent has been retained from the previous iterations of the study.

We find that a corresponding simplification only changes results by < 4% relative to the last round of findings. To prevent call-outs of oversimplification, and aligned with our intention of presenting readers with a general physics-based framework, the governing equation for climb energetics is retained, while stating that it can ultimately be used in the future if an accurate velocity profile and VTOL configuration is made available. In doing so, the overall integrity of our approach is maintained, questionable assumptions are avoided, and the key takeaways are found to be largely similar.

Change to Manuscript

Please see changes made corresponding to comment part 2 above.

Comment Part 4: Alternatively, energy used in the complex acceleration and deceleration segments could reasonably be assumed to be approximately covered by the present hover modeling (i.e., 30 seconds of steady hover at each end of the flight), and the climb and descent could be modeled as taking place at the cruise speed of 150 mph and its associated L/D of 17. However, the total energy usage calculated in such a way will not differ significantly from the total energy usage calculated with the previous suggestion of assuming only cruise power over the entire flight distance, so the trivial increase in fidelity may not be worth the additional modeling complexity. If the authors do prefer to extrapolate the cruise L/D curve to lower speeds, the Cirrus SR22 reference is not necessary, given the L/D figures already provided by the Uber reference. (The SR22 reference is also less relevant, since the Uber values are more representative of such a VTOL air taxi aircraft than those derived from the SR22). The widely-accepted two-parameter quadratic drag model can be developed for this aircraft from the two L/D values given in the Uber reference—17 at 150 mph and 13 at

200 mph—resulting in the curve $L/D = 1 / [(V / 790 \text{ mph})^2 + (22.6 \text{ mph} / V)^2]$, where V is the equivalent airspeed in mph.

Response

We appreciate the reviewer providing us with two possible pathways. We proceeded with using the prior alternative recommended by the reviewer. Please note that references to the Cirrus aircraft have also been subsequently deleted.

Central to our aim of providing readers with a general, high-level, physics-based framework for estimating VTOL energetics is retaining the model for climb. While we present rationale for us having adopted a simplified approach for modeling the climb phase, in an instance when readers have access to an accurate climb velocity and L/D , they can use our suggested equation to further refine their results. In this manner, we are also able to retain our consideration of a higher-fidelity climb model.

Change to Manuscript

Please see changes made corresponding to comment part 2 above.

REVIEWERS' COMMENTS:

Reviewer #2 (Remarks to the Author):

The authors have thoroughly responded to my comments on the previous draft. I recommend this article for publication.